

# Passive seismic experiment 'AniMaLS' in the Polish Sudetes (NE Variscides)

Monika Bociarska[1], Julia Rewers[1], Dariusz Wójcik[1], Weronika Materkowska[1], Piotr Środa[1] and *AniMaLS Working Group\**

[1]Department of Seismic Lithospheric Research, Institute of Geophysics Polish Academy of Sciences, Warszawa, 01-452, Poland

*A full list of authors and their affiliations appears at the end of the paper.

*Correspondence to*: Monika Bociarska (bociarska@igf.edu.pl)

**Abstract.** The paper presents information about the seismic experiment AniMaLS which aims to provide a new insight into

the crustal and upper mantle structure beneath the Polish Sudetes (NE margin of the Variscan orogen). The seismic array composed of 23 temporary broadband stations was operating continuously for ~2 years (October 2017 and October 2019). The dataset was complemented by records from 8 permanent stations located in the study area and in the vicinity. The stations were deployed with inter-station spacing of approximately 25-30 km. As a result, good quality recordings of local, regional and teleseismic events were obtained. We describe the aims and motivation of the project, the stations deployment

procedure, as well as the characteristics of the temporary seismic array and of the permanent stations. Furthermore, this paper includes a description of important issues like: data transmission set-up, status monitoring systems, data quality control, near-surface geological structure beneath stations and related site effects etc. Special attention was paid to verification of correct orientation of the sensors. The obtained data set will be modelled using several seismic interpretation methods, including analysis of seismic anisotropy parameters, with the objective of extending knowledge about the

lithospheric and sub-lithospheric structure and the tectonic evolution of the study area.

## 1 Introduction

The passive seismic experiment AniMaLS (**Ani**sotropy of the **Ma**ntle beneath the **L**ower **S**ilesia) aims at studying the structure of the crust and upper mantle of Polish Sudetes and Sudetic Foreland, as well as the processes of their orogenic evolution, using seismological and petrological methods. Up to now, the upper mantle in this region was only sparsely

sampled by seismic data (Wilde-Piórko et al., 1999; Wilde-Piórko et al., 2008). A temporary seismic array deployed in the Polish Sudetes in a period from October 2017 to October 2019 collected good quality seismic data, which are an important prerequisite to image the lithospheric and sub-lithospheric properties of Sudetes and the Lower Silesia region.

The main purpose of this paper is to present the research objectives of the AniMaLS project and technical information concerning the data acquisition. We describe the characteristics of the temporary seismic array and of the permanent stations





in the study area. Also, we present details of the stations deployment procedure, including the site selection, sensor orientation, data transmission set-up, status monitoring systems. We also discuss near-surface geological settings of the sites and their relation with observed site effects. We describe the technical aspects of field measurements, distribution and acquisition parameters of the stations, and stages of data quality control. Attention was paid to the verification of the sensor orientation. Finally, we describe the data completeness, present data examples, and discuss the noise characteristics.

The Lower Silesian region comprises two major tectonic units: the Sudetes mountains and the Sudetic Foreland, representing NE termination of the Central European Variscides. Its present shape is a result of a long and complex tectonic evolution. The Variscan consolidation of this region involved accretion of several Proterozoic and Palaeozoic, mostly Gondwana-derived terranes at the Laurussia margin. Present topography of Sudetes mountains is a result of Tertiary tectonic reactivation related to the Alpine orogeny. The project aims for a deeper understanding of the structure, tectonic evolution

and geodynamics of the NE Variscides using seismic methods, based on recordings of local, regional and teleseismic events. The depth range of the study comprises the crust and the mantle lithosphere, the lithosphere–asthenosphere boundary (LAB) and the sub-lithospheric upper mantle. Interpretation of the data with P- and S-receiver function will be attempted in order to trace the lithospheric and deeper discontinuities. Moreover, the project seeks to determine with more detail the structure of the mantle and its seismic anisotropy, using shear-wave splitting method applied to SKS and SKKS phases. Observations of

the anisotropy of the seismic wave velocity are an important tool for studies of the processes shaping the lithosphere. The character of the anisotropy reflects the degree and the direction of tectonic deformations of the lithosphere (or the orientation sub-lithospheric mantle flow) in the studied area. The analysis of the spatial distribution of seismic anisotropy will be attempted - potential differences in anisotropy parameters can be a proxy for discrimination between lithospheric blocks with different petrological composition or different tectonic evolution.

## 2 Station deployment

### 2.1 The network layout and equipment

The AniMaLS seismic array has been deployed between October 2017 and January 2018 and it operated for a period of ~2 years, until October 2019. Two institutions contributed to the temporary seismic network – the Institute of Geophysics, Polish Academy of Sciences (IG PAS) provided with 10 Guralp CMG-6T (equivalent of CMG-40T) 30s seismometers with

Guralp DM24S3EAM data acquisition units and one CMG-6TD 30s seismometer and recorder. The Institute of Geophysics of the University of Warsaw (IG UW) supplied 12 Reftek-130B recorders with broadband seismometers Reftek 151-120 "Observer" with bandwidth of 0.0083-50 Hz (120-0.02 s). Additionally, for observations of local seismicity, IG PAS deployed 6 units with short-period (1s corner frequency) Mark L-4C sensors. All stations had 130 dB dynamic range and used 100 Hz sampling frequency. Timing was provided by GPS. The average inter-station distance in the array was about

25-30 km.



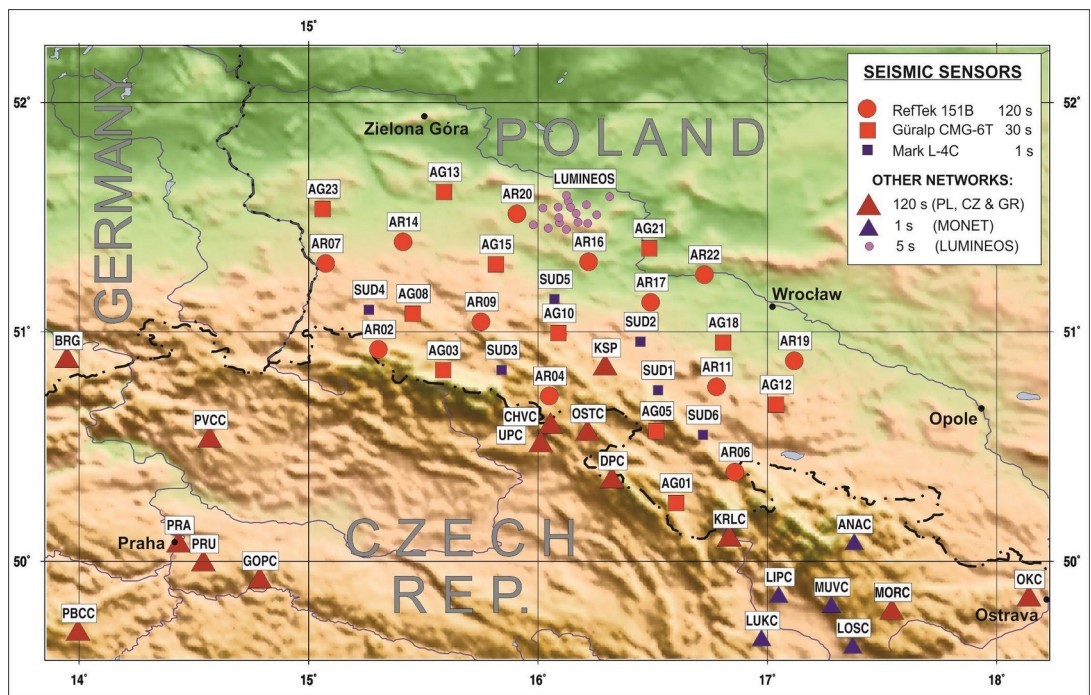

**Figure 1: Location map of the AniMaLS experiment. The red circles are the temporary broadband sites with 120 s sensors, the red squares – temporary sites with 30 s sensors, the dark-red triangles are permanent stations (120s sensors). Blue squares – short-period (1 s) temporary stations. Elevation map based on GTOPO30 dataset (U.S. Geological Survey, 1996).**


As several permanent seismic stations were operating in the study area, it was possible to enlarge the dataset with recordings of stations: KSP (Polish Seismological Network) and CHVC, DPC, KRLC, MORC, OKC, OSTC, and UPC (Czech Seismological Network), all equipped with 120 s sensors. The short-period (1s–5s sensors) LUMINEOS network, designed by IG PAS for monitoring of the induced seismicity in Legnica-Głogów Copper District (LGCD) (Mirek and Rudziński,
2017) is also in the area we are investigating. The distribution of the AniMaLS stations and permanent seismic stations used in the experiment is shown in detail in Fig.1. The coordinates of the stations, locations names and technical details are summarized in Table 1.

**Table 1. Location and technical parameters of the temporary and permanent stations used in the experiment, with lithology and**
**stratigraphy information for observation sites.**

| Station Code | Operation period | | Latitude [°] | Longitude [°] | Elev. [m] | Corner period [s] | Sample rate [Hz] | Sensor type | Site name | Lithology/stratigraphy at the surface |
|---|---|---|---|---|---|---|---|---|---|---|
| | **From** | **To** | | | | | | | | |
| **AG01** | 21-10-2017 | 23-10-2019 | 50.2540 | 16.6020 | 477 | 30 | 100 | CMG-40T | Ponikwa | limestones, marls (Upper Cretaceous) |
| **AG03** | 22-10-2017 | 23-10-2019 | 50.8329 | 15.5866 | 665 | 30 | 100 | CMG-40T | Piechowice | granitoids (Upper Carboniferous) |
| **AG05** | 20-10-2017 | 22-10-2019 | 50.5675 | 16.5159 | 535 | 30 | 100 | CMG-40T | Nowa Ruda | sandstones, mudstones (Carboniferous- |




| | | | | | | | | | | |
|---|---|---|---|---|---|---|---|---|---|---|
| | | | | | | | | | | Permian) |
| **AG08** | 23-10-2017 | 23-10-2019 | 51.0795 | 15.4545 | 387 | 30 | 100 | CMG-40T | Rząsiny | phyllites, shales (Lower Palaeozoic) |
| **AG10** | 22-10-2017 | 23-10-2019 | 50.9946 | 16.0892 | 315 | 30 | 100 | CMG-40T | Siedmica | greenstone shists, amphibolites (Lower Devonian) |
| **AG12** | 20-10-2017 | 22-10-2019 | 50.6827 | 17.0377 | 167 | 30 | 100 | CMG-40T | Witostowice | Quaternary clastics on Lower Palaeozoic |
| **AG13** | 16-11-2017 | 24-10-2019 | 51.6099 | 15.5909 | 145 | 30 | 100 | CMG-40T | Dzikowice | Quaternary clastics on Lower Palaeozoic |
| **AG15** | 23-10-2017 | 24-10-2019 | 51.2949 | 15.8169 | 152 | 30 | 100 | CMG-40T | Groble | Quaternary clastics on Lower Palaeozoic |
| **AG18** | 19-10-2017 | 22-10-2019 | 50.9533 | 16.8070 | 121 | 30 | 100 | CMG-40T | Kryształowice | Quaternary clastics on Lower Palaeozoic |
| **AG21** | 16-11-2017 | 25-10-2019 | 51.3643 | 16.4854 | 94 | 30 | 100 | CMG-40T | Tarchalice | Quaternary clastics on Mesozoic |
| **AG23** | 13-04-2018 | 16-10-2019 | 51.5358 | 15.0611 | 145 | 30 | 100 | CMG-6TD | Wymiarki | Quaternary clastics on Mesozoic |
| **AR02** | 29-12-2017 | 17-10-2019 | 50.9240 | 15.3048 | 550 | 120 | 100 | RT_151-120 | Świeradów Zdrój | schists, amphibolites (Upper Proterozoic – Lower Palaeozoic) |
| **AR04** | 29-11-2017 | 17-10-2019 | 50.7202 | 16.0492 | 484 | 120 | 100 | RT_151-120 | Lipienica | conglomerates, arkose sandstones, mudstones (Lower Permian) |
| **AR06** | 01-12-2017 | 18-10-2019 | 50.3894 | 16.8585 | 504 | 120 | 100 | RT_151-120 | Orłowiec | schists, amphibolites (Upper Proterozoic – Lower Palaeozoic) |
| **AR07** | 07-05-2018 | 26-03-2019 | 51.2981 | 15.0742 | 177 | 120 | 100 | RT_151-120 | Pieńsk | Quaternary clastics on Mesozoic |
| **AR09** | 29-11-2017 | 16-10-2019 | 51.0441 | 15.7518 | 342 | 120 | 100 | RT_151-120 | Bełczyna | conglomerates, arkose sandstones, mudstones (Lower Permian) |
| **AR11** | 14-12-2017 | 18-10-2019 | 50.7605 | 16.7783 | 215 | 120 | 100 | RT_151-120 | Ligota Wielka | gneisses, migmatites (Ordovician) |
| **AR14** | 17-11-2017 | 16-10-2019 | 51.3926 | 15.4122 | 145 | 120 | 100 | RT_151-120 | Ławszowa | Quaternary clastics on Lower Palaeozoic |
| **AR16** | 01-12-2017 | 26-08-2019 | 51.3060 | 16.2202 | 129 | 120 | 100 | RT_151-120 | Raszowa Mała | Quaternary clastics on Lower Palaeozoic |
| **AR17** | 14-11-2017 | 15-10-2019 | 51.1296 | 16.4912 | 141 | 120 | 100 | RT_151-120 | Wrocisławice | Quaternary clastics on Lower Palaeozoic |
| **AR19** | 13-11-2017 | 18-10-2019 | 50.8739 | 17.1170 | 153 | 120 | 100 | RT_151-120 | Kończyce | Quaternary clastics on Lower Palaeozoic |
| **AR20** | 15-11-2017 | 16-10-2019 | 51.5151 | 15.9078 | 135 | 120 | 100 | RT_151-120 | Nowa Kuźnia | Quaternary clastics on Lower Palaeozoic |
| **AR22** | 13-12-2017 | 15-10-2019 | 51.2493 | 16.7257 | 17 | 120 | 100 | RT_151-120 | Miękinia Głogi | Quaternary clastics on Lower Palaeozoic |
| **KSP** | 12-1999 | present | 50.8428 | 16.2931 | 353 | 120 | 20/100 | STS-2 | Książ | conglomerates, mudstones, limestones (Upper Devonian) |
| **CHVC** | 05-2009 | present | 50.5881 | 16.0547 | 580 | 120 | 20/100 | STS-2 | Chvaleč | carbonatic sandstones, arkose sandstones (Lower Permian) |
| **DPC** | 01-1993 | present | 50.3502 | 16.3222 | 748 | 120 | 20/100 | STS-1 | Dobruška/Polom | amphibolites, gabroamphibolites (Lower Palaeozoic) |
| **KRLC** | 11-2008 | present | 50.0966 | 16.8341 | 614 | 120 | 20/100 | CMG-3ESP | Králíky | gneisses (Lower Palaeozoic) |
| **MORC** | 05-1994 | present | 49.7768 | 17.5425 | 742 | 120 | 20/80 | STS-2 | Moravský Beroun | shales, mudstones (Lower Carboniferous) |
| **OKC** | 10-1998 | present | 49.8346 | 18.1399 | 250 | 120 | 20/100 | CMG-3ESP | Ostrava/ Krásné Pole | shales, mudstones (Lower Carboniferous) |
| **OSTC** | 10-2005 | present | 50.5565 | 16.2156 | 556 | 120 | 20/100 | STS-2.5 | Ostaš | marls, limestones (Upper Cretaceous) |
| **UPC** | 05-2001 | present | 50.5074 | 16.0121 | 416 | 120 | 20/100 | STS-2 | Úpice | dolomitic sandstones, arkose sandstones (Upper Carboniferous-Permian) |

## 2.2 Site selection and array design

The sites for permanent broad-band seismic stations are usually carefully selected in areas with extremely low noise. The sensors are located in vaults designed to minimize noise resulting from thermal and atmospheric variations. Such a careful site preparation and installation is often not possible in case of temporary seismic projects, where selection of the location, installation and formal issues (permissions, rental contracts) have to be done in a short time and with limited resources.






Additionally, to form a more or less uniform network, the sites should be located at similar inter-station distances, which is another constraint for the site location. When deploying the array, we attempted to obtain a compromise between several factors: low seismic noise, site availability, continuous power supply and good UMTS signal. An important issue was high
level of security, in order to avoid the damage or loss of the equipment. Meeting all these requirements was not straightforward, since in most of the locations the level of anthropogenic noise was elevated due to high population density and industrial activities. In these areas, fulfilling both constraints (stations spacing and low noise) has been extremely hard. When possible, we placed the units at the unused basements of buildings, in the outbuildings or in rarely used public utility buildings. The sensors were placed on a hard surface – concrete or tiled floor, and in some cases a 5 cm thick granite slab
was used for this.

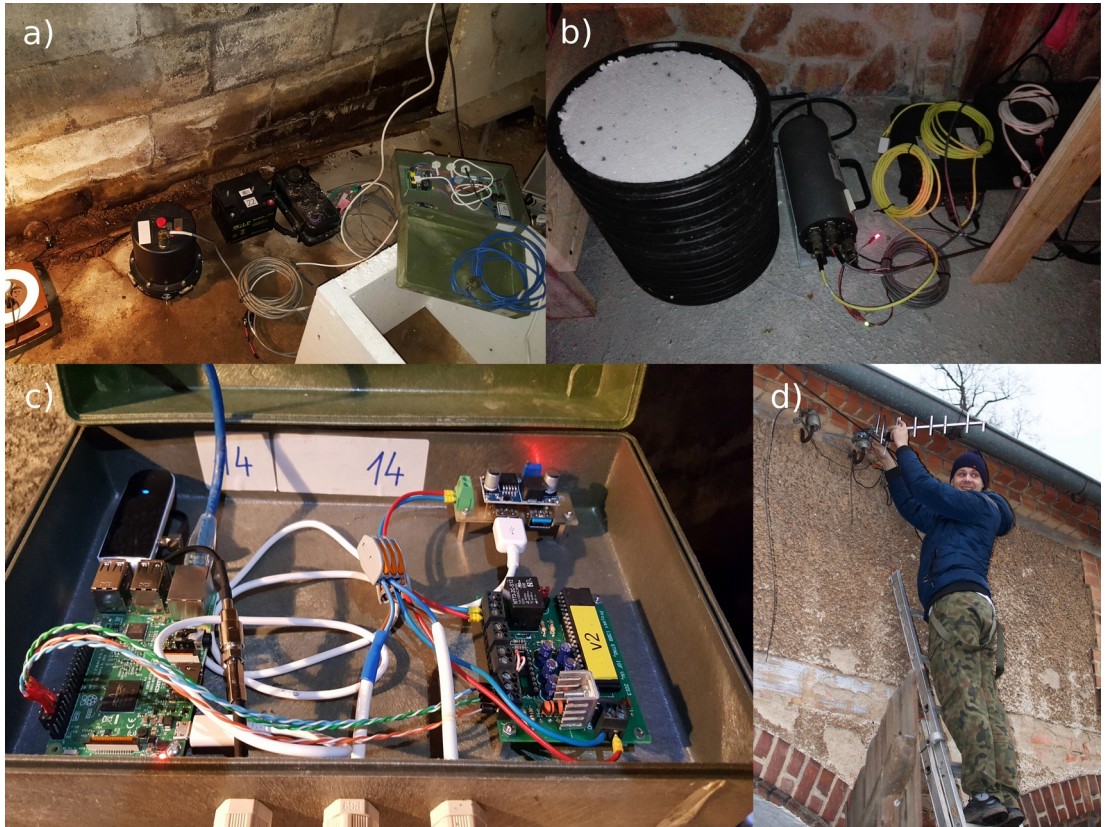

**Figure 2: Deployment of seismic stations in Sudetes: a) Reftek unit during installation, b) Installed Guralp unit, c) data-transmission module – Raspberry Pi microcomputer with UMTS modem and watchdog, d) |Installing the UMTS antenna for data transmission.**

At the sites, a thermal insulation of the sensor was ensured in a form of a stryrofoam box covering the sensor. A few pictures from installation of a typical station are shown in Fig. 2. Each station was powered by power grid system. The 12V power



supply was buffered with 40-60 Ah batteries in order to ensure continuous operation of the units in case of the power outages. Near real-time data transfer was done with UMTS/LTE mobile network connection.

## 2.3 Orientation of sensors

A precise orientation of seismic sensor axes with respect to geographical North direction is of great importance during installation of a 3-component seismic station. Incorrect orientation of the seismmometer can result in substantial errors when using 3-component methods of interpretation, e.g., in case of shear wave splitting analysis (Ekström and Busby, 2008; Vecsey et al., 2014; Wang et al., 2016). The simplest method of geographical North determination, using a magnetic compass, often results in uncertainty exceeding 5° (Vecsey et al., 2017), which is not satisfactory for some interpretation

methods. The modern approach, involving the use of an optical gyrocompass, allows for much higher precision, but requires expensive equipment.

Taking these limitations into account, we have designed our own low-cost system for precise orientation of the seismometers deployed in the project. For the determination of the geographical North direction in the field, we used GNSS RTK unit and ASG-EUPOS network (Ryczywolski et al., 2008) for receiving location corrections. Two ways for transferring the North

direction to the seismometer location was considered. First method was geodetic tachymetry. However, this method is not only time-consuming but also causes problems in less accessible locations such as basements. To solve this problem, we developed a simple device for azimuth transfer which makes the process more time-efficient while retaining satisfactory precision.

The core of the device is a MEMS triple axis accelerometer and gyroscope unit MPU-6050, controlled by a single-board

Raspberry Pi microcomputer. The data communication between device modules is based on I2C serial protocol over the General Purpose Input Output (GPIO) ports. The code for processing the data from the gyroscope unit was written as a Python script. Raw data from the unit are converted into stable values of the rotation angle of the device. Problem of the gyroscope drift was solved by calibration of the immobile device prior to the measurement phase. During calibration the drift is evaluated, and, based on this, corrections for drift are continuously applied during the measurement.

Orienting the seismometer towards the geographic North with this method is done in two stages. First, GNSS RTK unit is used to obtain precise positions of two points of a baseline outside of the station site and to calculate the azimuth of the baseline as a reference. Next, the azimuth is transferred to the place where the seismometer will be installed. To this end, the gyroscope device is aligned parallel to the baseline using laser pointer, and GNSS-measured reference azimuth value is input to the device. The device is then moved indoor to the station site where it is rotated to North, according to displayed current

azimuth, and the N-S line is marked on the floor at the location of the sensor. Finally, to check if the device readings were stable during the North measurement at the site, the device is moved back to the baseline and oriented along it, where, ideally, reference azimuth value should be again displayed. If the value differs substantially from the reference, it indicates excessive/variable drift or other errors, and the measurement is considered to be invalid. The procedure is repeated until 2-3



stable (negligible drift) and consistent measurements are obtained. Assuming availability of the GNSS RTK unit, this
method is an affordable solution which allows for orientation of the sensor with error determined in field tests to be of ±2
degree order.

## 2.4 Real-time data transmission and data storage

The seismic data were written in the internal storage of the recorders (Guralps - 16GB Flash memory, Refteks 2x16GB CF
cards) and, simultaneously, they were transmitted in near real-time to dedicated acquisition servers at the IG PAS.
Additionally, state of health (SOH) information including temperature, voltage, and mass positions were transmitted. The
data transmission was done using UMTS internet connection, with all devices running IPSec VPN system to securely
connect all the stations to the data acquisition server and to protect the system from unauthorized access. The Guralp units
were connected to the network using Mikrotik routers with LTE modems. The data transfer to a dedicated CMG-NAM data
hub was based on GDI protocol with a back-fill buffer, which allows for handling temporary loss of internet connection and
retransmission of missing data packets after the connection is re-established. Connection loss and router/modem hang-up
situations were handled by recorder unit by an implemented software watchdog, which allowed for 3 levels of action: soft
reset of the modem, power-cycling of the modem and power-cycling of the unit and of the modem. For data transmission
from Reftek units, a modified system designed at IG UW (Polkowski, 2016), based on Raspberry Pi Linux microcomputers
with UMTS/LTE wireless modems was used. The Raspberry Pi units served both as routers and as devices scheduling the
data transmission - collecting data from recorders and sending them to server (FTP, rsync and SSH protocols). The control
scripts (PHP, bash) were designed to check for gaps in transferred data (due to e.g., network connection loss, server or
device hang-up) and to schedule data retransmission, if necessary. Hardware watchdog devices, designed at IG UW
(Polkowski, 2016), were used to assure automatic restart of the transmitting unit on no connection or hang-up.

Both Guralp and Reftek stations were remotely controlled and monitored using their proprietary software providing a WWW
control interface. It allowed for checking the status of the individual units, mass positions, timing, voltages, temperature, as
well as setting various recording parameters. For Reftek units, the control interface alllowed for monitoring of mass
positions and mass centering. Also, automatic mass centering could be triggered if mass voltages exceeded a threshold after
a user-defined time.

The near real-time data transfer has well-known advantages - inspection of the current data flow and access to current SOH
information is useful for monitoring of the data quality and allows for fast detection of failures, such as power supply
malfunctions or timing problems. Also, an increase of the noise level or the signal distortion due to an inadvertent moving or
tilting of the sensor can be detected, and necessary station maintenance can be planned. This saves the number of the field
trips needed for servicing the stations and helps to quickly reestablish proper acquisition of the seismic data.

The gaps in transmitted data resulting from the lack of GSM connection were filled by periodic retrieving of the recorded
data directly from the recorder memory during stations maintenance in the field, if needed. The transmitted and patched data



were stored in miniSEED format. After unification of information in headers, the daily miniSEED files were finally stored in the form of SeisComP Data Structure (SDS) – a hierarchical structure with file and directory naming convention which allows for easy access to the data, e.g., with the ObsPy package.

## 2.5 Station timing

The seismic studies require exact measurement of absolute time of the seismogram to be able to determine the arrival times of the analysed phases. An incorrect timing may lead to erroneous identification of the phases or incorrect travel-time determination. Currently, the seismic recorders use GPS/GNSS receivers that allow the synchronization of the internal clock with a high accuracy (±10 µs). However, in practice, technical malfunctions or loss of GNSS signal can introduce timing errors, and such problems should be recognized. If possible, incorrect timing should be corrected during initial data
processing, or reported, to avoid using badly timed data for the interpretation. During the data acquisition for the project, an important problem with timing occurred for 5 Reftek recorders due to the "Week Number Roll-Over" (WNRO) issue in the GPS system in 2019, which affected the GPS receivers with older hardware, not designed to cope with this issue. As a result, in July 2019, some of the stations started to report the date with wrong year (e.g., 2099) and incorrect day of year. However, the correct time of the day was preserved, therefore it was easy to obtain proper date by shifting the time by a fixed amount
of full days. Corrected date/time was then written into miniSEED headers. Nevertheless, the wrong date caused malfunction of the online data transmission system, which expected a correct date in the transmitted file names and in the headers. The transmission system software had to be temporarily modified in order to avoid the problem. Permanent solution of the problem was later achieved by updating recorders' firmware with patched, WNRO-aware version.

Other problem was detected at the AG23 station, equipped with CMG-6TD recorder: after few weeks, the internal clock lost
synchronization with GPS time, in spite of properly working and locked GPS receiver. This resulted in a linear increase of the time difference, which reached ~20 s after few months of recording. In this case, only an approximate time correction was possible. By comparing the timing of good-quality arrivals in seismograms from AG23 and neighboring, correctly timed stations, it was possible to measure the time differences over the recording period, and to apply appropriate corrections. Here, the accuracy of time determination after the correction was estimated to ~1 s. This is a relatively large value, and it
prevents such data from being used for modelling methods which require exact knowledge of the absolute time, as e.g., seismic tomography. Nevertheless, such data can still be used in methods based on relative time of the seismogram components, as receiver function method or shear-wave splitting.

## 2.6 Characteristics of observation sites and near-surface geology

The geology of the near-surface sequences varies considerably over the study area, ranging from Proterozoic crystalline
rocks to unconsolidated Quaternary sequences. The geological structure of the basement at the observation site can heavily affect the character of the recorded seismograms, therefore we summarize the differences in the near-surface lithology and





discuss their possible influence on the seismic data. The Table 1 presents locations, technical information (sensor type, operation time), lithology and stratigraphy at the site for temporary and permanent stations. Geological information is based on the Geological Map of Poland 1:500000 (Państwowy Instytut Geologiczny - Państwowy Instytut Badawczy, 2021), and

Geological Map of Czech Republic 1:50000 (Geological map 1 : 50000, 2021).

In the SW part of the study area (roughly corresponding to the Sudetes mountains), the observation sites are located directly on consolidated rocks of Palaeozoic or Proterozoic basement (except AG01 and OSTC, positioned on Cretaceous rocks). The stations in the NE (the less-elevated region of Sudetic Foreland) are located on a layer of Quaternary and Tertiary unconsolidated sediments, overlying the Palaeozoic basement. This area is marked in Fig. 3 with green dotted line. The

presence of the low-velocity Cainozoic deposits at these sites has a distinct influence on the seismic records, and more detailed discussion of these effects is presented in Sect. 3.1.

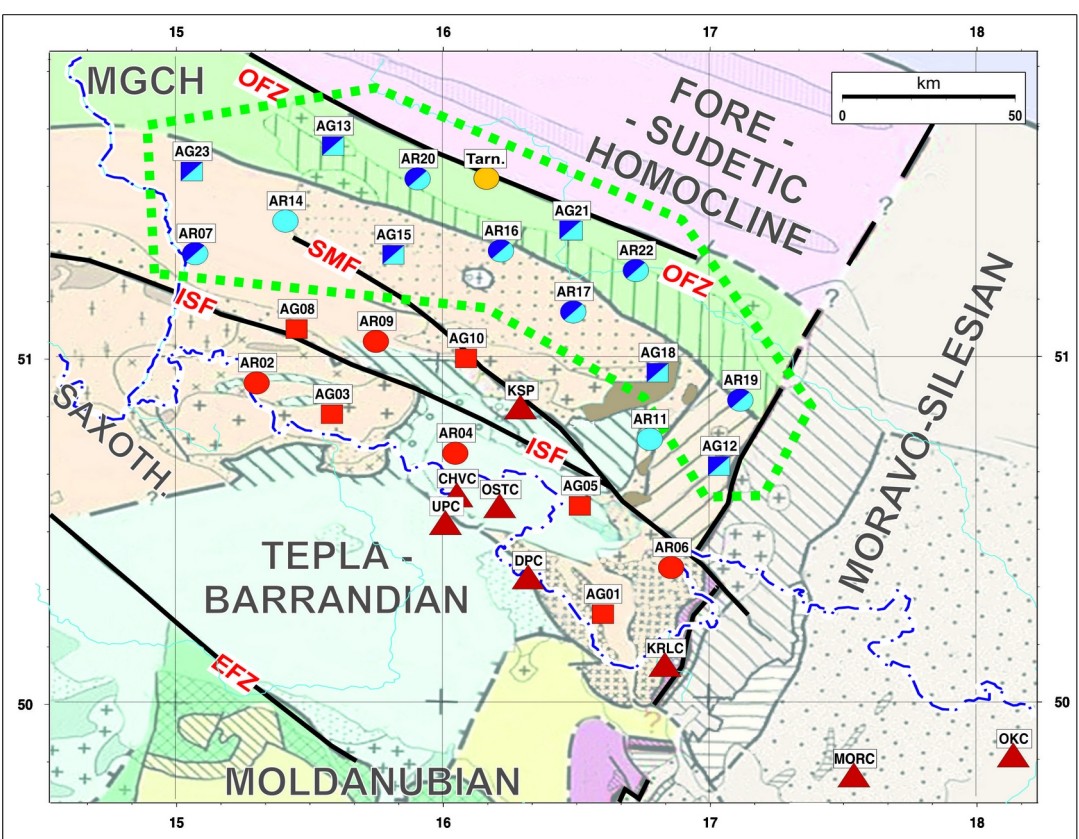

**Figure 3: Locations of temporary (circles and squares) and permanent (triangles) stations used in the experiment on a background of a tectonic map (modified after Franke et al., 2017). EFZ- Elbe Fault Zone, ISF- Intra-Sudetic Fault, MGCH – Mid-German**

**Crystalline High, OFZ- Odra Fault Zone, SMF – Sudetic Marginal Fault. Green dotted line delimits area where observation sites are located on Cainozoic sediments. Other stations are mostly located on Palaeozoic or Proterozoic basement, and two stations are on Cretaceous rocks. Light blue marks – stations with high noise amplitude in the short-period range, dark blue marks – stations showing high amplitude, long coda of the P-phase on horizontal components (see Sect. 3.1). Yellow circle - location of station in Tarnówek in LGCD (Mendecki et al., 2016), discussed in Sect. 3.1.**



## 3 Data

According to the ISC catalogue, during the registration period (October 2017 – October 2019) 1285 events with magnitude above 5.5 occurred. These earthquakes are shown in Fig. 4. Size of the dots represents the event magnitude. Figure 5 shows the data availability diagram for the stations of the array, produced using ObsPy package (Beyreuther et al., 2010). Several shorter gaps, mostly resulting from data transmission problems and some longer gaps (caused by hardware failures or power shortages due to heavy thunderstorms) are present. The overall completeness of the network-transmitted data, supplemented with untransmitted data after recovery in the field, is 97%.

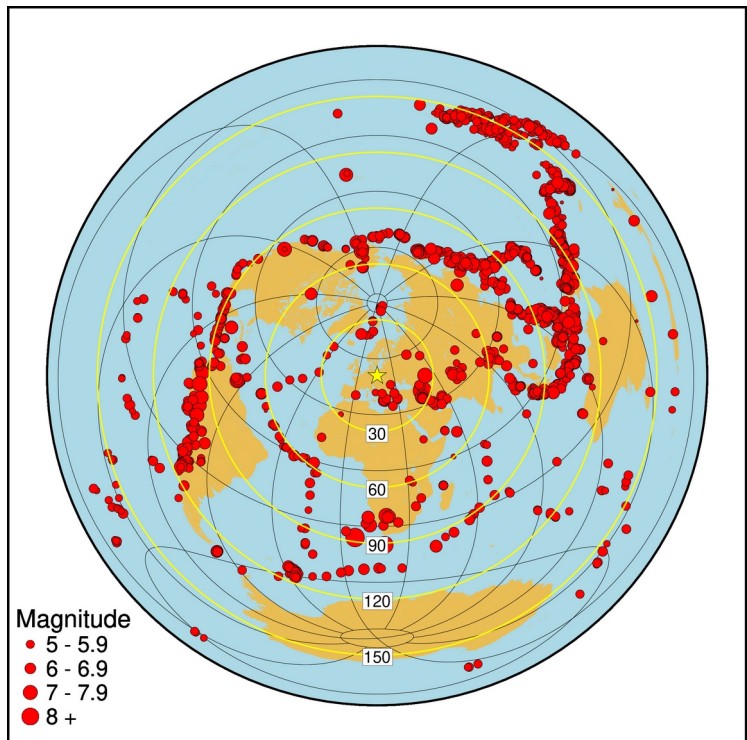

**Figure 4: Distribution of the epicenters of M > 5.5 earthquakes in the period from 18 Oct 2017 to 26 Oct 2019, according to ISC catalogue (1285 events). The yellow asterisk represents the center of the AniMaLS seismic array in Sudetes. The yellow circles mark the distances (in degrees) from the array center in 30° steps.**





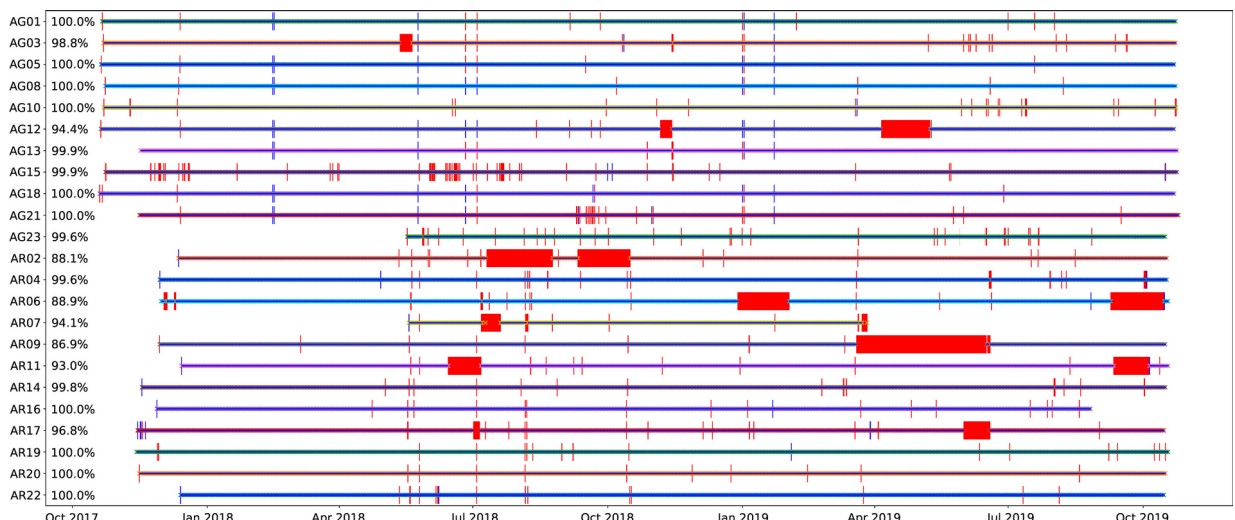

**Figure 5: The diagram showing the data completeness for temporary stations. Red fragments – gaps in the data resulting from stations failures, memory cards errors, power shortages.**

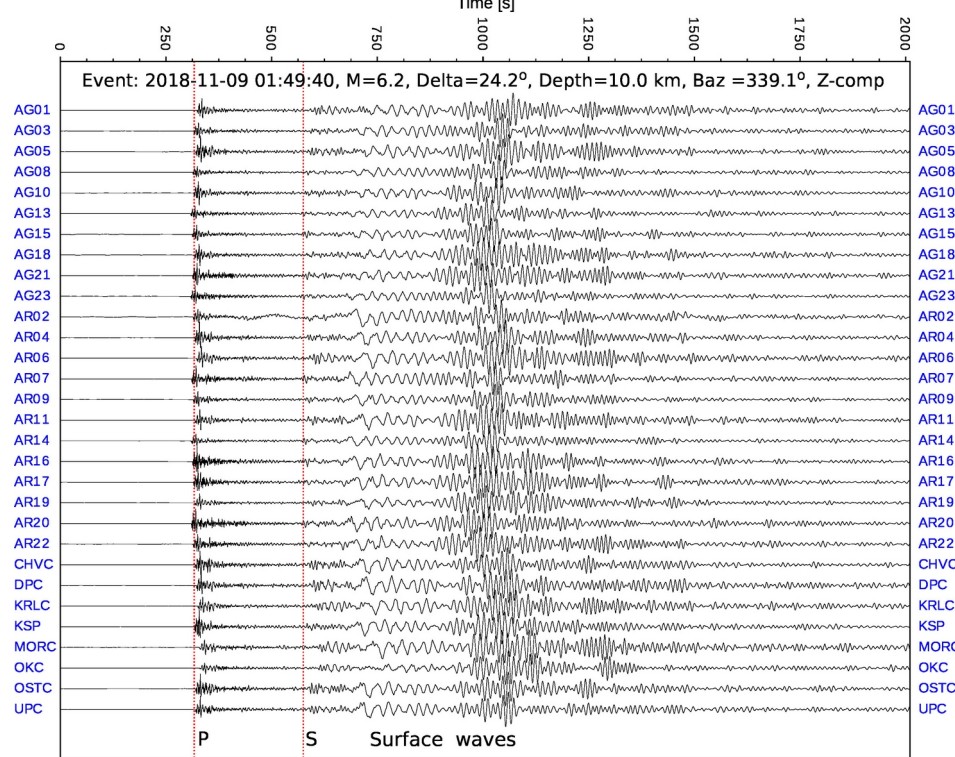

**Figure 6: Example of vertical component of recorded waveforms for the M6.2 teleseismic earthquake which occurred 2018-11-09 in Jan Mayen Island region. Red lines mark the theoretical onsets of P- and S-phases at the KSP station. All seismograms are low-pass filtered ( < 1 Hz).**




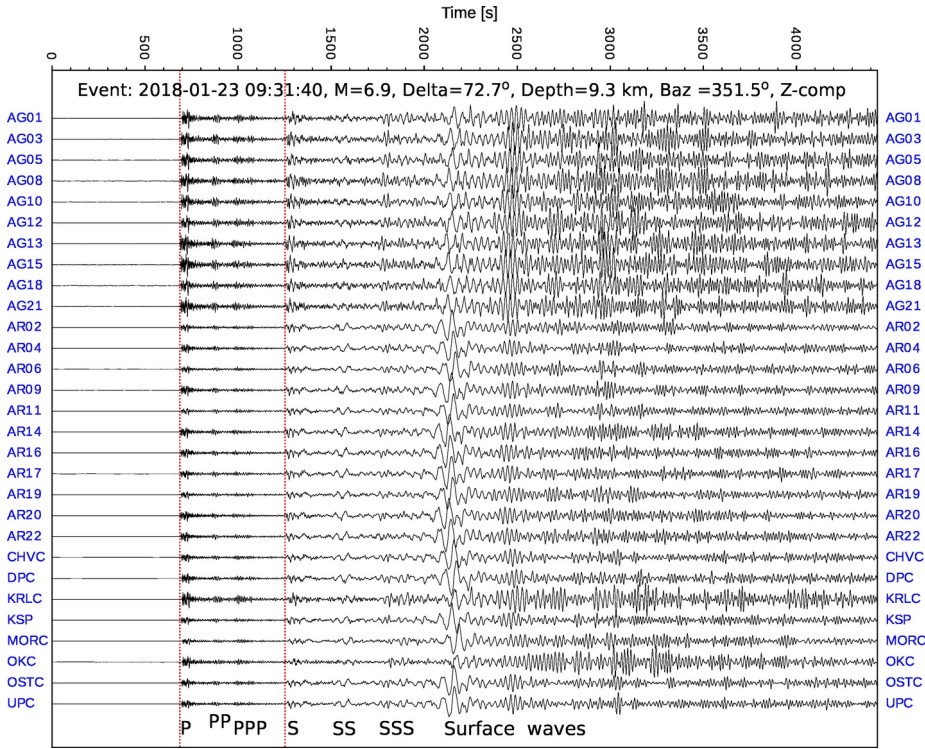

**Figure 7: Example of vertical component of waveforms recorded for the M6.9 teleseismic earthquake which occurred 2018-01-23 near Alaska. Red lines mark the theoretical onsets of P- and S-phases at the KSP station. All seismograms are low-pass filtered ( < 1 Hz).**

Figure 6 presents an example of seismograms for an earthquake near Jan Mayen Island with M = 6.2, at epicentral distance Δ = 24°. It occurred 2018-11-09, 01:49:40.05 UTC (lat: 71.6312, lon: -11.2431, depth: 10.0 km after ISC). The red dotted lines mark the theoretical arrival times of P- and S-waves at the KSP station. The seismograms show strong P-wave arrivals and lower-amplitude S arrivals, followed by high-amplitude surface (LR) waves, showing distinct dispersion. Figure 7 shows an example of a teleseismic earthquake from Alaska area (M = 6.9, Δ = 73°), which occurred 2018-01-23, 09:31:40.91 UTC

(lat: 55.9315, lon: -149.1877, depth: 9.3 km after ISC). Here, besides high-amplitude P- and S-waves, also free-surface reflections PP, PPP, SS, and SSS can be clearly observed. Starting from ~2050 s relative time, a long train of surface waves with substantial dispersion is visible. Interestingly, this figure clearly shows differences in frequency response of sensors for two groups of stations (it should be noted that records are scaled to maximum amplitude of each seismogram). For the AR- and permanent stations, equipped with 120 s sensors, the strongest amplitude is seen for the earliest, long-period (~50 s)

pulses of surface wave at ~2050-2150 s time. However, for AG- stations, these long-period pulses are outside the 30 s corner frequency of the sensors and are strongly attenuated. With maximum trace amplitude scaling applied, this leads to substantial enhancement of amplitudes of remaining parts of the seismogram: the body-wave pulses and later surface wave trains (with periods < ~30 s) for AG- stations, relative to AR- and permanent stations records.




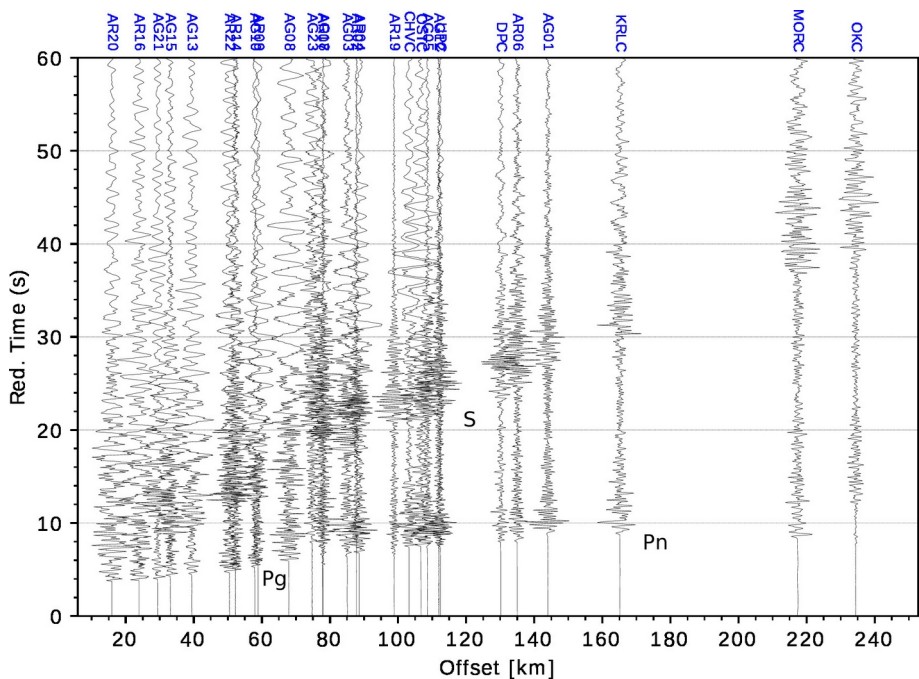

**Figure 8. Example of waveforms recorded for the M4.4 local earthquake which occurred 2018-07-03 in Legnica-Głogów Copper District. Band-pass filter 0.2-15 Hz was used. Reduction velocity is 8 km/s.**

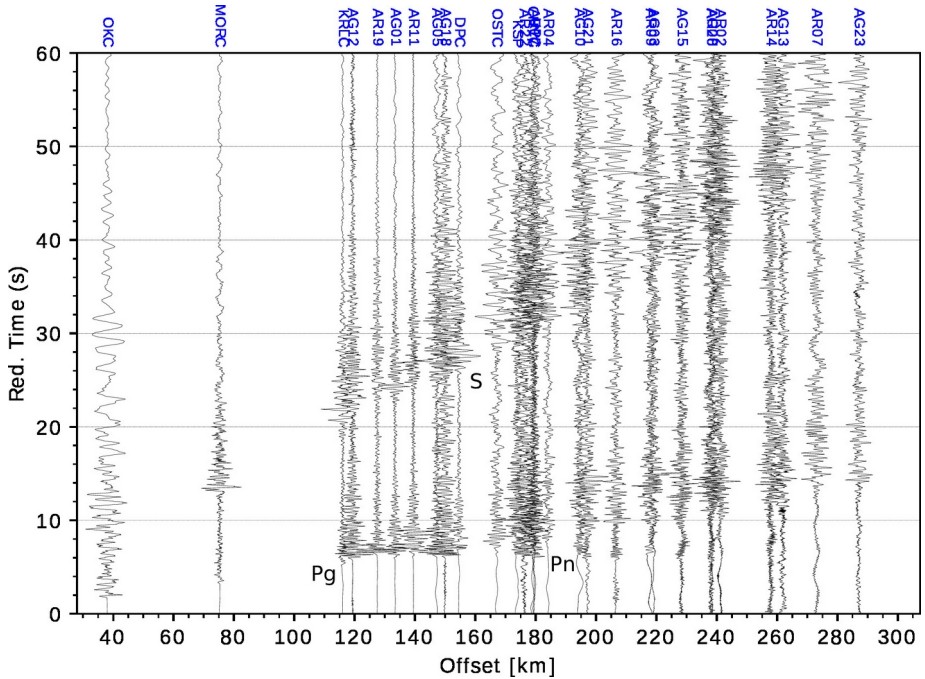

**Figure 9: Example of waveforms recorded for the M3.9 local earthquake which occurred 2019-01-22 in Upper Silesia district. Band-pass filter 0.2-15 Hz was used. Reduction velocity is 8 km/s.**



Figures 8 and 9 show local earthquakes from Legnica-Głogów Copper District and from Upper Silesia district, respectively.
Both events are related to local mining activities. The first event with magnitude 4.4 occurred 2018-07-03, 19:38:47.75 UTC
(lat: 51.5145, lon: 16.1378, depth: 0.0 km after ISC), the second one, with magnitude 3.9, occurred 2019-01-22, 22:35:30.08
UTC (lat: 50.1095, lon: 18.4559, depth: 5.5 km after ISC). These figures show the Z-component with 0.2-15 Hz bandpass
filter. The epicentral distances are in the range of 0-240 km for Legnica-Głogów event and 40-300 km for Upper Silesia
event. At these distances, we observe crustal phases - strong Pg phase, and mantle refraction – the Pn phase in the first
arrivals. The Pn appears at offsets > ~140 km. At larger times, strong S-waves and surface waves are recorded.



**Figure 10: Spectral seismograms obtained using Continuous Wavelet Transform for station AR09, showing a teleseismic event from Southern Alaska, 2018-11-30, 17:29:26 UTC. From top to bottom: E-, N- and Z-component.**



An example of time-frequency representation of the data is shown in Fig. 10. Here, a spectral seismogram obtained using
continuous wavelet transform (Daubechies, 1992) is presented for a teleseismic event (Southern Alaska, epicentral distance
67°, backazimuth 353°) recorded by AR09 station. The Morlet wavelet was used. The onset of the P-wave is visible at ~600
s relative time, in records of Z- and N-component, with maximum amplitude in 2-4 s period range. At ~1200 s travel time,
the S-waves with periods in 12-15 s range are visible, with the largest amplitude on E- component. The body waves are
much weaker than surface waves, which are visible at larger travel times. On the E-component, corresponding approximately
to transverse direction relative to ray backazimuth, the LQ waves with a period of 50-60 s can be seen at ~1600 s time. The
LR waves, best visible on N-and Z-component records at ~1900 s, clearly show the dispersion, with period changing from 40
s to 25 s.

## 3.1 Seismic noise characteristics and site effect

To estimate the level of the ambient noise at various frequencies, we calculated the probabilistic power spectral density
(PPSD) distributions (McNamara and Buland, 2004) for the data recorded at each station using ObsPy package (Beyreuther
et al., 2010). The PPSDs was calculated using continuous recordings from the period 01.12.2018-01.10.2019 (22 months).
The PPSD calculation was based on analysis of 1-hour-long windows of continuous seismic data (with 0.5 h overlap). The
processing sequence consisted of demeaning, tapering, FFT computation and instrument response removal. The obtained
frequency spectra for all windows were smoothed and summed to form a histogram representing the frequency distribution
of noise amplitudes at various period ranges. The result shows which amplitudes are observed for a given period. The PPSD
medians were also calculated.
Figure 11 shows a comparison of the PPSDs for three types of stations: AG10 (with 30s CMG-40T sensor), AR06 (RT 151-
120s sensor) and permanent station UPC (STS-2 sensor). Diagrams for three components are presented. Figure 12 shows the
PPSDs of Z-component for 12 selected temporary and permanent stations used in this study. Figure 13 shows a comparison
of PPSD median curves for all sites used, including permanent and temporary stations. There is a systematic difference in the
noise level between permanent and temporary sites. The difference is notable for long-period range (> 10 s), and is particu-
larly large for the horizontal components. High amplitude of the noise for the long periods of the horizontal components is
often experienced in case of temporary stations, mainly due to an imperfect protection from environmental thermal/pressure
changes or the sensor base tilt (Wilson et al., 2002). Another factor contributing to higher long-period amplitudes on the hor-
izontal component with respect to vertical amplitudes, in particular for stations located on young/low velocity sediments,
could be the ellipticity of the Rayleigh waves. In presence of a low-velocity layer, the Rayleigh waves exhibit horizontally
flattened particle motion, whereas at hard-rock sites on consolidated/crystalline basement, the particle motion is vertically
elongated (Tanimoto et al., 2013). However, here, this factor seems to have a minor influence, considering relatively small
(< 1 km) thickness of the low-velocity layer in this area, which should not affect the ellipticity of long-period (> 10 s) waves
in question.



**Figure 11: Probabilistic power spectral density (PPSD) for stations AG10 (CMG-40T), AR06 (RT-151-120) and permanent station UPC (STS-2). The Z, N and E components at the top, middle and bottom, respectively. The time span for calculation is 22 months from Jan 2018 to Oct 2019. Black lines mark New High and Low Noise Models (NHNM, NLNM; Peterson 1993).**





**Figure 12: Probabilistic power spectral density (PPSD) on the Z-component for 12 selected stations. Black lines mark New High and Low Noise Models (NHNM, NLNM; Peterson 1993).**



The highest amplitude of the long-period noise, often exceeding New High Noise Model (NHNM) level, characterizes all sites with 30 s CMG-40T sensors. Previous studies show similar behaviour of these sensors, independently of the actual noise at the site (Evangelidis and Melis, 2012; Custodio et al., 2014, Staehler et al., 2016). This is most likely due to high self-noise of this device type, and, partially (for $T_{noise} > 30$ s), to lower corner period of the CMG-40T device (30 s vs. 120 s

for other units).

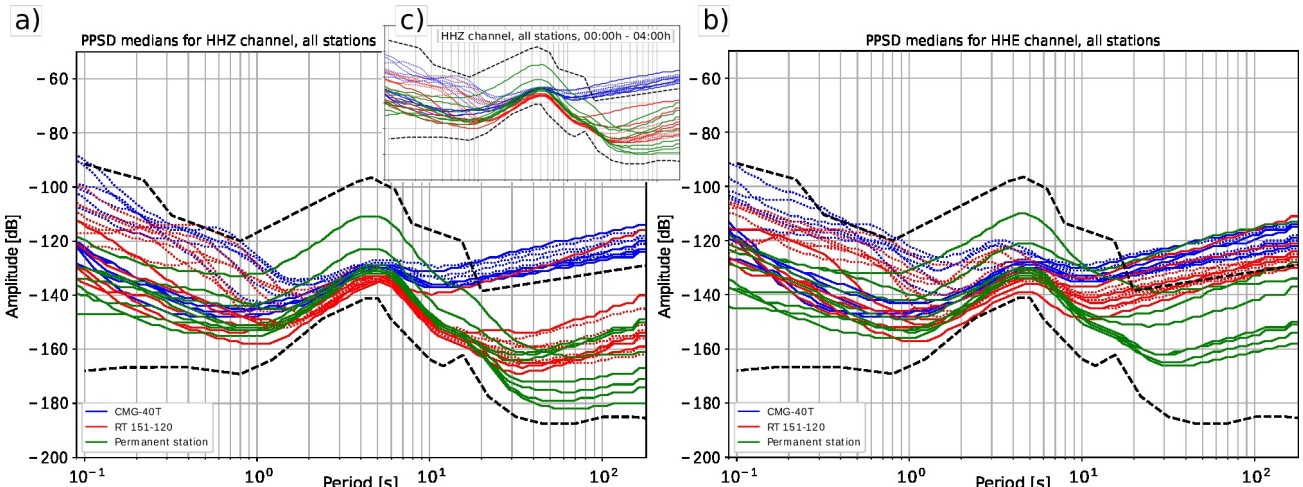

**Figure 13: The PPSD median curves for all temporary and permanent stations. a) Z-component, b) E-component, c) Z-component, night hours only (00:00-04:00 local time). Dotted lines – stations located on Quaternary sediments, solid lines - stations located on Palaeozoic or older basement. Time span for calculation is 22 months from Jan 2018 to Oct 2019. Black lines mark New High and**
**Low Noise Models (NHNM, NLNM; Peterson 1993).**

The short-period parts of the all PPSD medians (Fig. 13) show amplitude differences independent of the station type, and can be subdivided into two groups. Stations located on Palaeozoic or older, consolidated basement (solid lines in Fig. 12) show much lower noise in this part of the spectrum than the stations on the basement covered by unconsolidated, alluvial Quater-
nary/Tertiary sequences (dotted lines). This area represents NE part of the array, marked with green dotted line in Fig. 3. The stations with high amplitude of the short-period noise, marked with light blue color, mostly fit into this region, which suggests a high correlation of this effect with the basement type. When attempting to interpret these differences in terms of the near-surface geology, care must be taken, because the high-noise sites installed on the Quaternary cover are, in the same time, located in the area with higher population density, denser network of roads, expressways and railroads, with typically
higher anthropogenic noise. To eliminate or minimize the influence of the cultural effects, the PPSD medians were calculated also for the same time span (22 months) but only for night hours – from 00:00h to 04:00h every day. The result (Fig. 13c) still shows clear difference in the short-period noise amplituse between the sites located on old Palaeozoic rocks and the sites on the young Quaternary cover.



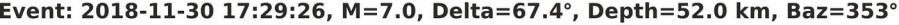

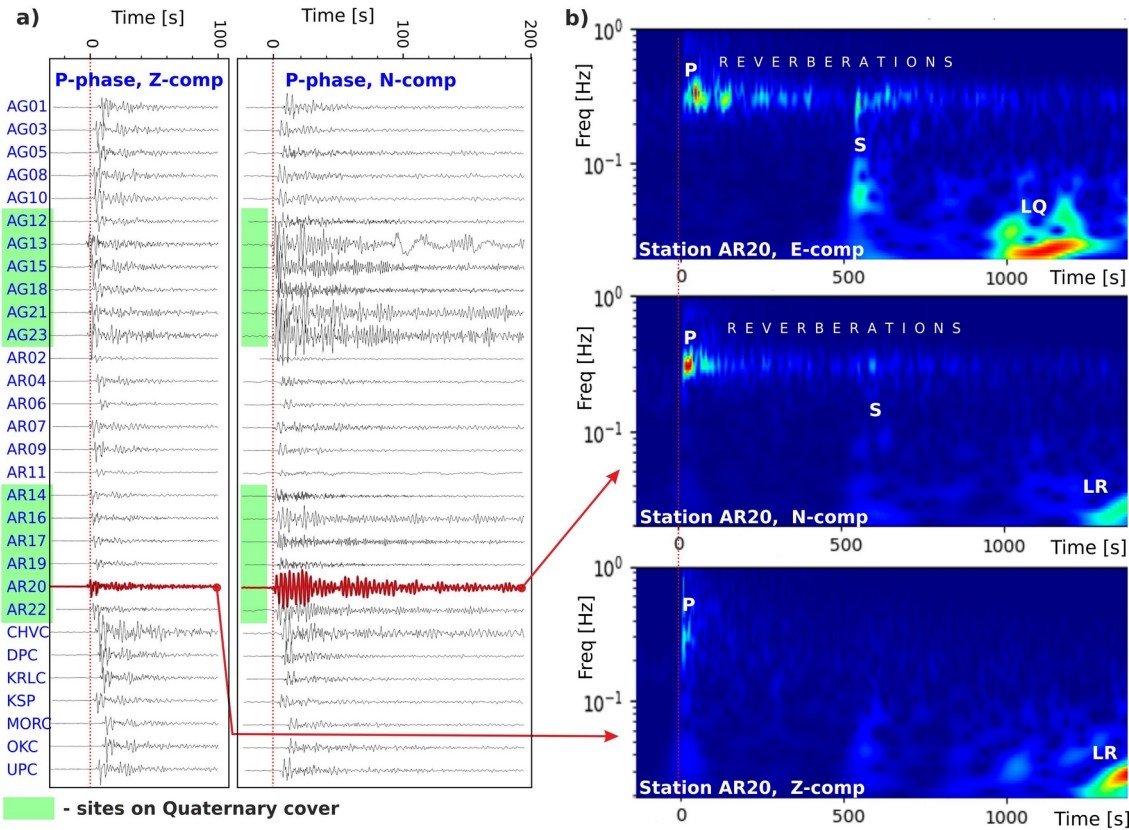

Figure 14: Seismic data example for event from 2018-11-30, 17:29:26 UTC, illustrating the differences in P-phase records between stations located on consolidated basement (short pulse, no reverberations) and sites located on young, unconsolidated cover (green rectangles, strong reverberations on the horizontal components). a) Z- and N-component records for all the stations used. b) Spectral 3-component seismograms for station AG20. E- and N-components show high amplitude, > 600 s long coda (reverberations) in a narrow frequency range, centered at 0.3 Hz. The coda is non-existent in the Z-component record. True amplitude scaling was applied. Time scale is relative to theoretical P-phase onset.

The presence of the low-velocity sediments in the area of Sudetic Foreland is also related to another effect, affecting the character of the P-phase onsets. The P-wave pulses on the horizontal components are followed by a prominent, high-amplitude coda/reverberations, extending over up to several hundreds of seconds (Fig. 14). The coda is characterized by a narrow frequency range, with a central frequency of 0.25 - 0.40 Hz (periods of 2.5 - 4 s), depending on the station location. The corresponding P pulses on the vertical component are much shorter and seem to be only weakly affected (or not affected) by the coda. In contrast, for the stations located on the consolidated basement such reverberations are not observed on any component (Fig. 14a).

Such phenomenon was long recognized and described by several authors, e.g. by Zelt and Ellis (1999) or Yu et al. (2015), as it may heavily distort the results of 3-component interpretation methods. A layer of low-velocity sediments, with a strong impedance contrast relative to the consolidated or crystalline basement, produces multiple P-to-S conversions and reflections





between the free surface and the sediments bottom. This results in high-amplitude reverberations in a narrow frequency range, mostly visible on the horizontal components. The frequency of the multiples is directly related to the seismic velocity and the thickness of the low-velocity layer. A systematic determination of the properties of the near-surface layer is out of scope of this paper. However, these observations can be compared with studies of the NE part of the study area (LGCD),

where the properties of the low-velocity layer were studied by Mendecki et al. (2016). They used the HVSR method to ana-lyse the resonance frequencies and amplification factors based on the data collected by a broadband station in Tarnówek (Fig. 3), located ~15 km to the East of AR20 station. The HVSR peaks at 3.6 -4.2 s were found and Vs of ~0.4 km/s was es-timated for a ~380 m thick Cenozoic layer at this location. In our study, the Fig. 14b shows shorter (3.3 s) main period of the coda for AR20 station, which most likely correspond to thinner sedimentary layer or higher S-wave velocity.

The reverberations related to a low-velocity layer pose significant problems for the interpretation of the data, e.g., with the receiver function technique, as they overprint Ps conversion pulses on the radial component. One of the methods to overcome this problem was presented by Yu et al. (2015). As the reverberations exhibit a resonant frequency related to the two-way traveltime of the wave in the sediment layer, the approach is based on designing a resonance removal filter in the frequency domain with filter parameters derived from the properties of the autocorrelation of the calculated RF. Our first

tests showed that such filter, applied to the data from Sudetic Foreland, is quite effective and significantly reduces the effect of reverberations.

### 3.2 Verification of sensors misorientation

In the case of seismological methods which require three-component seismic recordings, such as receiver function analysis or shear-wave splitting method, precise sensor orientation is crucial to get correct results. Significant misorientation of the

seismometers can adversely affect the reliability of final results and the interpretation. During the installation of stations in the field, to assure correct orientation, an azimuth measurement system with GNSS RTK unit and a MEMS gyroscope was used, as described in Sect. 2.3. According to our estimates, such system allows for determination of the N direction at the sensor location with ±2° accuracy, if appropriate care is taken by the operator during all steps of the procedure. In order to additionally check for possible misorientation of the sensors after deployment, using the acquired data, a method based on

the analysis of the P-wave polarization described by Fontaine et al. (2009) was applied. These estimates were verified using a method proposed by Braunmiller et al. (2020), based on the P-wave polarization, and with approach of Doran and Laske (2017) based on polarization of the Rayleigh waves. The two latter methods are implemented in the OrientPy package (Audet, 2020).

Assuming a homogeneous and isotropic medium, in a seismogram recorded by a correctly oriented sensor, the polarization

of the P-wave and of the Rayleigh wave particle motion is expected to be confined to the ray plane, and its horizontal component to be polarized_parallel to the event backazimuth. The misorientation of the seismometer (deviation of the N seismometer axis from the geographical North by *A* degrees - equivalent to rotation of the coordinate frame of the



measurement system) will obviously result in an apparent deviation of polarization of the P-wave from the ray direction by an angle -A, independently of the event backazimuth. However, in real medium, this deviation can be superimposed by the effects of the heterogeneity (dipping velocity discontinuities) or anisotropy of the medium under the station: (Crampin et al., 1982; Schulte-Pelkum et al., 2001; Fontaine et al., 2009). These effects show a specific azimuthal dependence of resulting deviation angles (periodic with 180° or 360° period), therefore it is often possible to separate these factors, if data from a wide range of backazimuths are available. The total directional variability of the polarization deviation can be decomposed as (Schulte-Pelkum et al., 2001):

$$D_{pol}(\alpha) = A + B\sin(2\alpha) + C\cos(2\alpha) + D\sin(\alpha) + E\cos(\alpha),\tag{1}$$

where particular terms reflect the magnitude of various factors: A – the constant (azimuth-independent) component of polarization deviation, directly related to the incorrect sensor orientation; B and C – effect of anisotropy with horizontal symmetry axis; D and E – effect of anisotropy with inclined axis or effect of an inclined discontinuity; $\alpha$– the event backazimuth.

For the analysis, from 165 events in the epicentral distance range of 5°-100°, the recordings with high signal-to-noise ratio of the P-phase (SNR > 5) were selected for each station. Selected data were filtered (various sub-bands of 2-16 s period band were used) and 3-D particle motion at the P-onset was analysed using the orthogonal distance regression (ODR) method implemented in the ObsPy package, providing the azimuthal angle of the motion in the horizontal plane and incidence angle. Also, rectilinearity as defined by Fontaine et al. (2009) was calculated and was used to reject arrivals with poor rectilinearity of the particle motion, as contaminated by noise or other effects, and likely to produce distorted results. The error of the azimuthal angle was determined based on calculated eigenvalues of the particle motion (Fontaine et al., 2009). In order to improve stability of final results, individual $D_{pol}$ values were sorted into backazimuthal bins of 30° width and averaged. Subsequently, these mean values were used for fitting the curve based on the equation (1) and for calculation of A-E parameters. The constant parameter A corresponds to the sensor misorientation.

To verify the results, we also analysed the same data set with a recently released software package OrientPy (Audet, 2020). The package implements two methods of determination of sensor orientation. The method described by Braunmiller et al. (2020) (BNG) determines the direction of P-wave polarization by minimizing the energy on the transverse component in a selected window around P-wave onset (Wang et al., 2016). Subsequently, polarizations for all events are averaged. The averaged value represents the constant component of the azimuth-dependent deviations, and is related to the misorientation angle for given station. It should be noted that the BNG method relies on relatively uniform backazimuthal coverage of the analysed data – averaging of a non-uniformly sampled sinusoidal curve is likely to result in a biased estimate of the mean value. Obtaining the mean value A by fitting the curve (X) to the points as proposed by Fontaine et al. (2009) should produce more reliable result if the azimuthal distribution of the data is highly inhomogeneous.

The Doran and Laske (2017) method (DL) is based on Rayleigh-wave polarization analysis. For each event, a search is done for an angle $\alpha$ (defined relatively to theoretical backazimuth) which maximizes the cross-correlation between the Hilbert





transform of the vertical component and the radial component rotated by $\alpha$. As in the BNG method, calculated individual deviations for all the analysed events are averaged to get a value of the misorientation of individual stations.

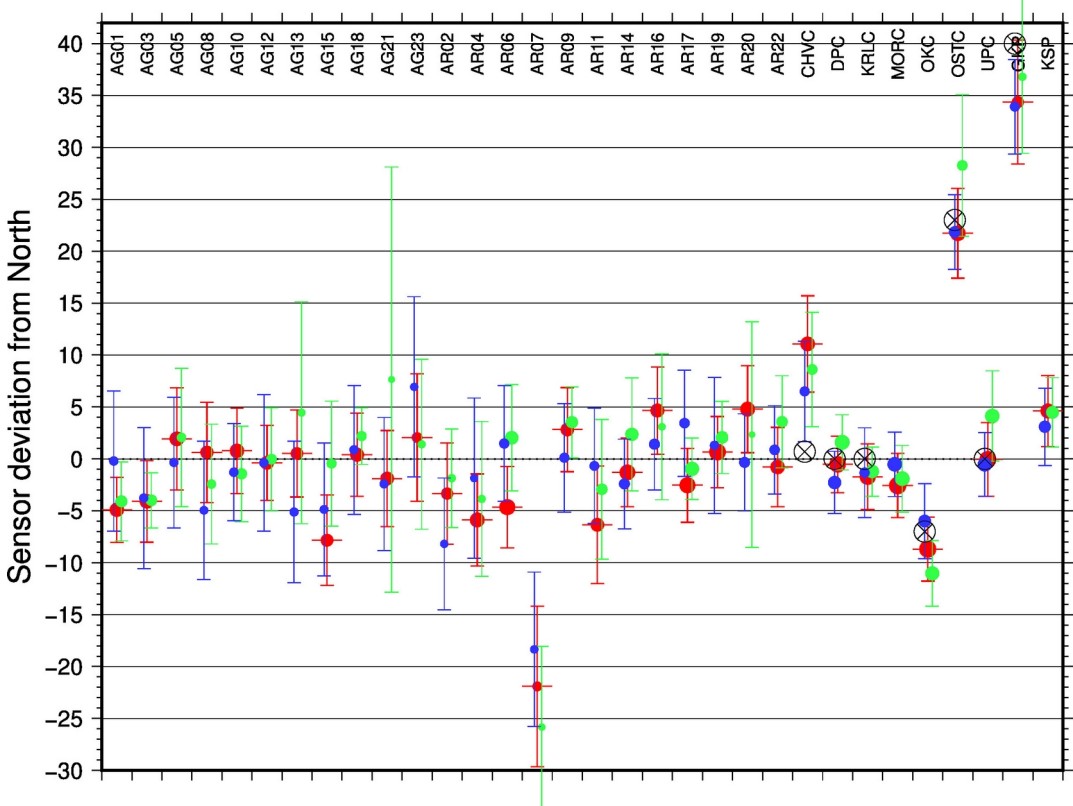

**Figure 15: Misorientation angles calculated for all stations using three described methods: red dots – Fontaine et al. (2009) method, green dots – BNG method, blue dots – DL method. Black circles with crosses – orientation angles of permanent stations measured using direct (gyroscope) methods in field (Vecsey L., Institute of Geophysics of Czech Academy of Sciences, personal communication, 2020) or (for GKP station) with indirect methods.**

Figure 15 shows obtained values of misorientations of all stations in the study area using three described methods (the permanent station GKP is outside the study area, but it is shown for comparison, as previous studies also reported its significant misorientation - Vecsey et al. (2014) reported 41°, Wilde-Piórko et al. (2017) reported 39° and 45°, result of this study: 34-37°. The measured sensor misorientation is significant for AR07, OSTC and GKP stations, with absolute values in a range of 20°-37°. For most other stations, the values do not exceed the range -7° to +7°. For many stations, the results derived from the three methods are more or less consistent, but with some conspicuous exceptions. It can result from a small amount of recordings used for analysis because of low SNR for several stations.

For some permanent stations of the Czech seismological network (CHVC, DPC, KRLC, OKC, OSTC and UPC ) the orientation angles obtained from direct, high accuracy measurements in field were available (Vecsey L., Institute of Geophysics of Czech Academy of Sciences, personal communication, 2020). They are presented as a reference in Fig. 15.



For almost all these stations (except CHVC) our results are in a good agreement (in a ± 2-3° range) to the direct measurements.


**Table 2. Misorientation angles for all stations, obtained from three methods, and from direct measurements in field, if available. (*) not a direct measurement, result of Wilde-Piórko et al. (2017). (**) not a direct measurement, result of Vecsey at al., (2014).**

| Station Code | Fontaine (2009) method | BNG method | DL method | Direct measurement |
|---|---|---|---|---|
| AG01 | -4.90 ± 3.14 | -4.09 ± 3.80 | -0.21 ± 6.76 | -- |
| AG03 | -4.09 ± 3.94 | -3.99 ± 2.66 | -3.78 ± 6.80 | -- |
| AG05 | 1.93 ± 4.93 | 2.06 ± 6.65 | -0.35 ± 6.30 | -- |
| AG08 | 0.63 ± 4.84 | -2.43+/-5.77 | -4.95 ± 6.67 | -- |
| AG10 | 0.78 ± 4.11 | -1.44 ± 4.60 | -1.28 ± 4.68 | -- |
| AG12 | -0.38 ± 3.60 | -0.04 ± 4.97 | -0.38 ± 6.57 | -- |
| AG13 | 0.53 ± 4.19 | 4.45 ± 10.69 | -5.11 ± 6.82 | -- |
| AG15 | -7.83 ± 4.35 | -0.45 ± 6.01 | -4.86 ± 6.41 | -- |
| AG18 | 0.41 ± 4.01 | 2.20 ± 2.73 | 0.87 ± 6.20 | -- |
| AG21 | -1.89 ± 4.63 | 7.64 ± 20.48 | -2.41 ± 6.41 | -- |
| AG23 | 2.06 ± 6.13 | 1.41 ± 8.17 | 6.94 ± 8.68 | -- |
| AR02 | -3.34 ± 4.86 | -1.86 ± 4.76 | -8.18 ± 6.34 | -- |
| AR04 | -5.88 ± 4.43 | -3.85 ± 7.45 | -1.85 ± 7.72 | -- |
| AR06 | -4.64 ± 3.91 | 2.04 ± 5.12 | 1.48 ± 5.57 | -- |
| AR07 | -21.91 ± 7.72 | -25.82 ± 7.77 | -18.34 ± 7.44 | -- |
| AR09 | 2.82 ± 4.05 | 3.53 ± 3.42 | 0.11 ± 5.23 | -- |
| AR11 | -6.35 ± 5.66 | -2.93 ± 6.72 | -0.67 ± 5.56 | -- |
| AR14 | -1.30 ± 3.29 | 2.36 ± 5.45 | -2.41 ± 4.32 | -- |
| AR16 | 4.66 ± 4.20 | 3.09 ± 7.02 | 1.42 ± 4.40 | -- |
| AR17 | -2.54 ± 3.56 | -0.96 ± 2.96 | 3.44 ± 5.11 | -- |
| AR19 | 0.66 ± 3.42 | 2.07 ± 3.47 | 1.30 ± 6.55 | -- |
| AR20 | 4.79 ± 4.19 | 2.34 ± 10.85 | -0.34 ± 4.68 | -- |
| AR22 | -0.78 ± 3.83 | 3.57 ± 4.43 | 0.87 ± 4.26 | -- |
| KSP | 4.62 ± 3.42 | 4.49 ± 3.34 | 3.09 ± 3.72 | -- |
| GKP | 34.36 ± 5.97 | 36.80 ± 7.37 | 33.92 ± 4.55 | 39 ± 2 (*), 45 ± 4 (*) 41 (**) |
| CHVC | 11.07 ± 4.64 | 8.61 ± 5.50 | 6.50 ± 4.82 | 0.70 |
| DPC | -0.52 ± 2.72 | 1.60 ± 2.66 | -2.27 ± 3.00 | 0.00 |
| KRLC | -1.72 ± 3.16 | -1.22 ± 2.38 | -1.33 ± 4.33 | 0.00 |
| MORC | -2.57 ± 3.09 | -1.91 ± 3.21 | -0.52 ± 3.10 | -- |
| OKC | -8.68 ± 3.09 | -11.03 ± 3.16 | -5.99 ± 3.62 | -7.00 |
| OSTC | 21.73 ± 4.33 | 28.26 ± 6.82 | 21.84 ± 3.59 | 23.00 |
| UPC | -0.05 ± 3.55 | 4.13 ± 4.35 | -0.53 ± 3.06 | 0.00 |



It must be noted that the results of the indirect, polarization-based, methods are not as precise as direct orientation
measurements, e.g., with the optical gyrocompass. According to Rueda and Mezcua (2015), the Rayleigh wave polarization
method achieves 1–5° uncertainty in case for long time spans of observations, e.g., at permanent stations, while for shorter
time intervals the uncertainty can exceed 10°. Therefore, as pointed out by Vecsey et al. (2017), in case of temporary arrays
with limited period of data acquisition, the methods based on polarization analysis are able to detect only substantial (>
~10°) misorientation of seismometers. Consequently, for our data, we decided that only results documenting misorientation
above 10° were meaningful, and only for these stations seismograms will be corrected by an appropriate rotation. The values
of the misorientation calculated by three methods for all the stations are summarized in the Table 2.

## 4 Conclusions and perspective

The AniMaLS project is an experimental seismic study of the physical properties and geological structure of the lithosphere
and sub-lithospheric mantle beneath the Polish Sudetes (NE margin of the Variscan orogen), with a complex history of
tectonic evolution. The acquisition of the seismic data involved deployment of 23 broadband stations for the period of ~2
years (Oct 2017 – Oct 2019). The selection of sites and installation was done using a low-cost approach, with the stations
deployed inside the unused basements, sheds or in rarely used public utility buildings. The stations were powered through
the power grid, and the data were collected using near real-time data transmission over the UMTS network. During the
measurement period, over 97% of data were retrieved. Location of the sites in the vicinity of the inhabited areas increased
the safety, the ease of installation and the reliability of the data transmission, however, at the cost of the noise level, which
was higher compared to the permanent stations in the region. Overall, the installed array provided a reliable acquisition of
the continuous broadband seismic data of good quality in near real-time. The acquired records of local, regional and
teleseismic events will be used for various seismic interpretation methods.

Analysis of P-receiver functions will allow for imaging of the lithospheric discontinuities, in particular of the Moho bound-
ary, and for comparison of the obtained Moho depths with the results from the wide-angle data modelling, e.g., from the
SUDETES 2003 seismic experiment (Grad et al., 2003) or based on other active seismic studies. Also, deeper mantle (410
km and 660 km) discontinuities will be imaged. Based on the S-receiver functions and on the analysis of surface waves, a
study of the lithosphere-asthenosphere boundary depth and properties will be attempted. The upper mantle anisotropy will be
studied using shear-wave splitting analysis, based on the recordings of the SKS and SKKS phases. The analysis of the P-
wave polarization may also contribute to anisotropy studies. The research on the anisotropy and its variations can provide the
information on the direction and degree of frozen-in lithospheric deformations or on the asthenospheric mantle flow, as well
as on the petrological characteristics of the mantle. The mantle studies will benefit from a good knowledge of the crustal
structure beneath the Sudetes, obtained based on previous active source experiments and geological information from
drillings.



In the past, the study area was covered by a single Polish seismic station KSP only, therefore the newly obtained recordings of the local events, acquired by the AniMaLS array, can be also of a great help for studies of the local seismicity, seismotec-tonics and in seismic hazard assessment.

Obtained geophysical results will be integrated with other geophysical and geological data. For instance, recent results of the petrological studies of anisotropy of the mantle xenoliths in the Tertiary volcanics (Puziewicz et al., 2012), abundant in
Sudetes, will be used to complement the seismic results and to help to interpret the origins of the mantle anisotropy. A multidisciplinary synthesis involving the results of the seismic interpretation can serve as a basis for inferences about relative movements of the tectonic units forming the area, about the impact of orogenic and other deformational events on the present structure, and can help to reconstruct the history of geological evolution of the NE Variscan orogen and of the neighboring areas.


*Data availability:* Data from the AniMaLS experiment are stored at the IG PAS (https://dataportal.igf.edu.pl/dataset/animals), currently with restricted access.

*Author contributions:* Article preparation was done by MB and PŚ with contributions from all co-authors. MB participated in
data acquisition, QC, processing and PPSD analysis. JR participated in data acquisition, QC, processing and analysis of stations misorientation. DW designed the azimuth-transfer system and participated in data acquisition. WM participated in data acquisition, QC and processing. PŚ designed the experiment, directed and participated in data acquisition, QC, processing and analyses.

*Team list:* The AniMaLS Working Group comprises: Marek Grad[1], Tomasz Janik[2], Kuan-Yu Ke[2,3], Marcin Polkowski[1] and Monika Wilde-Piórko[1,4].

*1 Faculty of Physics, Institute of Geophysics, University of Warsaw, Warszawa, 02-093, Poland*

*2 Department of Seismic Lithospheric Research, Institute of Geophysics Polish Academy of Sciences, Warszawa, 01-452, Poland*

*3 Helmholtz Centre Potsdam, GFZ German Research Centre for Geosciences, Telegrafenberg, 14473, Potsdam, Germany*

*4 Institute of Geodesy and Cartography, Warszawa, 02-679, Poland*

*Competing interests:* The authors declare that they have no conflilct of interests.

**Acknowledgements**

The project was funded by the National Science Center (grant no. UMO-2016/23/B/ST10/03204). Some of the figures were
created with Generic Mapping Tool software (Wessel and Smith, 1995). We thank people participating in preparation and





deployment of the seismic stations: T. Arant, M. Chmielewski, E. Gaczyński, J. Grzyb, S. Oryński, T. Skrzynik, J. Suchcicki. Special thanks for J. Wiszniowski for numerous valuable advices.

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
