# Peer review of "Passive seismic experiment 'AniMaLS' in the Polish Sudetes (NE Variscides)"

_Geoscientific Instrumentation, Methods and Data Systems, 2021_

## Referee Comment (RC2)

Review of ms. **Passive seismic experiment 'AniMaLS' in the Polish Sudetes (NE Variscides)** by M. Bociarska et al.

Submitted ms. documents in details recent passive seismic experiment AniMaLS in Polish Sudetes, both the station installation, network running and data collection, as well as data quality check. Though many described features are well known to seismologists operating temporary networks and working with their data, an advantage of the presented ms. consists in the documentation of the AniMaLS netwok. It can serve, among another "technical strategies" as a guide to students or less experienced young scientists in the field.

The ms. is well written in a simple style, though a language improvement worth for considering. I suggest several modifications of the ms. before considering it for publications.

The first general comment concerns the Introduction section. The section is too long and does not relate to main body of the technical description of the array and data. Moreover, a figure with tectonics of the region (including terms used in the text) and its location within Europe (an inset) are missing in this part. Not everybody is familiar with tectonics of the region. Part starting in Line 35 contains a lot of tectonic statements taken from literature, but without a single reference. This is unacceptable. Moreover, the Introduction should mentioned previous studies devoted to the crust (a series of active experiments) or concerning to the mantle lithosphere (passive experiments), incl. anisotropy and LAB, sub-lithospheric upper mantle. This is not the first study of this kind in the region as it is presented here (e.g., Majdanski et al., 2014; Grad et al., 2008; Karousova et al., 2012; Geissler et al., 2012, Plomerova et al., 2012; Vecey et al., 2014; Knapmeyer-Endrun etal., 2013; Kind et al., 2017; Kvapil et al., 2021), regardless of the fact that the main purpose of the ms. lies in the AniMaLS array documentation.

The second general comment relates to the repeating of too many numbers within the text, which are also in Figures, their captions and Tables. The continuous repeating makes the text chaotic and boring. Suppressing repetitions of several kinds will make the text shorter and clearly oriented on the technical aspects of the array.

What is the reason for plotting in Fig. 1 also stations not used in the study and the figure at all? Figure 3 would be more appropriate at the beginning of the ms. According to which criteria some of permanent stations are listed in Table 1 (station used in the study, as seems form further parts in the text) and some not (see Fig. 1). The Table 1 can contain, e.g., the institutions operating/owning the stations instead in the long text.

Not all abbreviations used in the text are introduced.

Line 18: strange formulation, usually data are used to model some feature (structure) …. not data Reformulate please.

Line 39: should be …aims at

Line 67, 415: should be ….. Czech Regional Seismic Network

Line 179 (e.g.): consider to use …. data acquisition system…. insted of …..recorder

Line 200: Cenozoic

Line 211: Sentence reformulation can help. Change order in the sentence, add (Fig.4) into it and delete the following two short sentences. Info in the second one is in the legend of the figure, there is no need to repeat it.  Check the ms. for this kind of its improvement. It can be done in many places.

Figures 6 and 7 are examples of regional and teleseismic earthquakes recorded by the network. Most information is repeated three times - in the figures, captions and text.  Clean the text (line 230 and further) from numbers, keep them in figure headings/legends. Location of time scale at the top of figures with an arbitrary origin is strange.

Line 245: adding location of the two local earthquakes into the Fig. 3 (even if it becomes Fig.1) would bring new information

Line 250: Do not repeat numbers presented elsewhere......The first event with magnitude 4.4 occurred 2018-07-03, 19:38:47.75 UTC .....

Line 259  or caption of Fig. 10 and many other places in the text: who is using? Instead consider ... with the use of ....

PPSDs shown in Fig. 11 do not support the frequent statements on "high-quality data" from the AniMaLS array. AG10 does not meet the High and Low-noise criteria neither on the Z component. Horizontal components of AR06 are poor. Splitting of the PPSD at UPS (permanent) comes from the winter increase of the micro-seisms.  Showing PPSD for all three components of the all temporary stations (in an electronic attachment) would be better than Figure 12 with selected temporary stations and permanent stations, whose PPSD are better. Is the effect of instrument type stronger than that of sediments at a site? It looks like that from Fig. 13. Please, comment on that. (Repetition of colour description in the legend and caption of Fig.13.)

Line 315:  cultural effects – anthropogenic effects are meant (not necessarily "cultural")

Line 321: 3-component spectral seismograms

Line 333: Such phenomenon is well known for a long time and described …..

Line 340 and around: too many abbreviations

Line 344:  ..... corresponds to a thinning of the sedimentary layer, or,....

Line 356:  repeats ...

Line 364: change order in the sentence

Line 405 and Figure 15: change label of GKP to be readable; put all labels ABOVE the frame; KEEP standard notation, i.e., that sensor mis-orientations greater than 5° prevents naming the horizontal components as N and E!!!, but the component must be denoted as Z, 1 (for N) and 2 (for E). This criterion should be accepted and followed (vs. Lines 412 and 430).  Wrong orientation of the GKP

station has been identified during processing data from the PASSEQ experiment and published in Vecsey et al. (2014)! However, presenting this information in the ms. is not correct, or, written in confusing way. What do you mean under the direct measurements in the field? In this form I understand that you mean measurements by gyrocompass. However, the GKP mis-orientation in Vecsey et al. (2014) was determined by software methods, as described there and mentioned in the Figure 15 captions (add ref Vecsey et al., 2014 for GKP deviation) but wrongly in the rightmost column of Table 2. I suggest:

Table 2. **Misorientation angles for all stations by different methods.**

| Station Code | Fontaine (2009) | BNG method | DL method | other / gyrocompass |
|---|---|---|---|---|
| **AG01** .................. | | | | |
| **GKP** | 34. 36+/- | 36. 80+/- □ | 33. 92+/- □ | 39+/- (Wilde-Piorko etal, 2017) 45+/- (Wilde-Piorko etal, 2017) 41 (Vecsey et al., 2014) |

**Followed by as you have it**
**CHVC** 11. 07 □} 4. 64 8. 61 □} 5. 50 6. 50 □} 4. 82 0. 70
**DPC** -0.52 □} 2.72 1.60 □} 2.66 -2.27 □} 3.00 0.00
**KRLC** -1.72 □} 3.16 -1.22 □} 2.38 -1.33 □} 4.33 0.00
**MORC** -2.57 □} 3.09 -1.91 □} 3.21 -0.52 □} 3.10 --
**OKC** -8.68 □} 3.09 -11.03 □} 3.16 -5.99 □} 3.62 -7.00
**OSTC** 21.73 □} 4.33 28.26 □} 6.82 21.84 □} 3.59 23.00
**UPC** -0.05 □} 3.55 4.13 □} 4.35 -0.53 □} 3.06 0.00

Line 416: Modify the sentence ……from direct, high accuracy measurements in field by gyrocompass (Vecsey L., Institute of Geophysics of Czech Academy of Sciences, personal communication, 2020).

Line 453: correct the sentence and add references for active experiments.

Line 455: In the past, the study area was covered by a single Polish seismic station KSP only …… This is incorrect statement. See, e.g, Fig. 2 in Vecsey et al., 2014 for PASSEQ experiment and other experiments covering the Sudetes (BOHEMA II) and related papers.

Line 465: Data availability: Data from the AniMaLS experiment are stored at the IG PAS (https://dataportal.igf.edu.pl/dataset/animals), currently with restricted access.

Do you plane to store the data in an EIDA node and when the data will be opened for public access?

---

## Referee Comment (RC3)

[referee-annotated manuscript omitted]

---

## Author Comment (AC1)

**Dear Dr. Stähler,**

**We are grateful to have received your review of our manuscript which led to important improvements of the manuscript. Below we address your comments in detail. Our responses are written in bold font.**

The manuscript *Passive seismic experiment 'AniMaLS' in the Polish Sudetes (NE Variscides)* by *Bociarska, Rewers et al* is a description of the installation process and data quality of a temporary network in Silesia, Southern Poland. The experiment is a high-density network of broadband, wideband and short period seismic sensors with a scientific focus on regional crustal structure and anisotropy.

The paper is well-written and complete and provides a full description of station installation and performance that will be helpful for future users of this dataset. It covers all the necessary parts of such a network paper and can be published with a few small revisions, mainly to improve readability.

L. 41: What is "the study"? Maybe write "the experiment" or "the network"

**Authors: OK, changed.**

L. 44: "Observations of anisotropy of seismic wave velocity" (remove the "the")

**A: OK, corrected.**

L. 49: Here I miss an overview over the paper. This description is about the network and the research planned with it. Could you please add a paragraph describing the structure of the paper?

**A: OK. Some short form of an overview was already present at L. 29-35 (original ms.). We moved it to the end of the chapter, extended it and made it more systematic.**

L. 52. Remove "it"

**A: OK, corrected.**

L. 54: remove "with" before 10

**A: OK, corrected.**

L. 130: Maybe mention here that a data-based verification was done and is shown in sect. 3.2?

**A: OK, explanation added.**

L. 164: I think that the official ObsPy reference is now Krischer et al 2015

**A: OK, corrected.**

L. 298: Please write as Stähler, in Latex St\"{a}her

**A: OK. We are sorry for incorrect spelling, we corrected it.**

L. 298 and figs 11, 13: I am surprised to see this low performance of a "normal" CMG-40T. So far, I had assumed that it was an issue with the OBS variant. The authors might want to reference

TasiÄ I., and Runovc F.; Seismometer selfâ noise estimation using a single reference instrument, J. Seismol 2012. 16, no. 2, 183–194, doi: https://doi.org/10.1007/s10950-011-9257-4 which shows a much better CMG-40T performance and

Simon C. Stähler, Mechita C. Schmidtâ Aursch, Gerrit Hein, Robert Mars; A Selfâ Noise Model for the German DEPAS OBS Pool. Seismological Research Letters 2018;; 89 (5): 1838–1845. doi: https://doi.org/10.1785/0220180056
Where we have a direct comparison of the "classical" CMG-40T and the OBS version. The noise curve shown here looks very much like what we saw for the "OBS version". This does not speak well for the manufacturer.

**A: OK. Yes, thank you for references, we will use them. Also, we realized and corrected our mistake which made the discussion and comparison of the sensors misleading:**

**We introduced some confusion in the manuscript due to incorrect naming of the sensors. The sensors we were using are actually CMG-6T, as we stated once at the beginning (in L.54 of the orig. manuscript), but we also (incorrectly) wrote that they were equivalent of CMG-40T sensor, and subsequently, we used this name in the discussion and figures. We did it because:**

**(1) We've been told by a Guralp representative that these are basically the same sensors from the user's point of view,**

**(2) the nominal responses of both sensors (based on IRIS NRL RESP files) were practically identical (differences in poles definitions at $4^{th}$ significant digit),**

**(3) technical specifications of both sensors, published by Guralp, were basically the same (except different sets of sensitivity options and operating temperature) – bandwidth, electronics noise level (-172 dB), power consumption (480 mW), dimensions and weight were exactly the same,**

**(4) initially, we could not find papers discussing the performance and PPSDs for the CMG-6T, but I found some for CMG-40T.**

**So, based on the above info, we assumed (wrongly) that, most likely, 6T and 40T are two commercial names for the same product, and I assumed (wrongly) that it is safe to refer to papers describing the performance of CMG-40T and compare them to our sensors.**

**Other thing is – we were not aware that two different versions of CMG-40T exist – land version and OBS version.**

**Therefore, your comment makes the situation more clear. We changed the text (and labels in figures), we used actual names of sensors (CMG-6T), corrected the discussion accordingly, and used the references you suggested. Also, after a more thorough search, we found one publication where PPSDs of CMG-6Ts were presented (Tillmann, 2006) and were consistent with our observations.**

**So, actually, there is no contradiction which surprised you – about land CMG-40T behaving as OBS CMG-40T, but, I guess it is still surprising why CMG-6T land sensors show the same/similar noise performance as CMG-40T OBS sensors.**

Figure 15: Please add a legend to the figure.

**A: OK, a legend added.**

Table 2: I think that two significant digits would be enough, given the sigma

**A: We rounded the numbers to one decimal digit, we hope it is acceptable.**

L. 445 -464: I think it is not really necessary to repeat all the scientific plans here, given that this paper is well-focused on the instrumentation

**A: OK, scientific plans are now shortened substantially, some parts of the original text were used in the Introduction.**

L. 466: Could you mention whether there is a plan for future public release of the data and metadata?

**A: OK, we added information that the data will be open in 2023.**

**A: OK, scientific plans are now shortened substantially, some parts of the original text were used in the Introduction.**

---

## Author Comment (AC2)

**Dear Anonymous Referee,**

**We are grateful to have received your review of our manuscript which led to important improvements of the manuscript. Below we address your comments in detail. Our responses are written in bold font.**

Review of ms. **Passive seismic experiment 'AniMaLS' in the Polish Sudetes (NE Variscides)** by M. Bociarska et al.

Submitted ms. documents in details recent passive seismic experiment AniMaLS in Polish Sudetes, both the station installation, network running and data collection, as well as data quality check. Though many described features are well known to seismologists operating temporary networks and working with their data, an advantage of the presented ms. consists in the documentation of the AniMaLS netwok. It can serve, among another "technical strategies" as a guide to students or less experienced young scientists in the field.

The ms. is well written in a simple style, though a language improvement worth for considering. I suggest several modifications of the ms. before considering it for publications.

The first general comment concerns the Introduction section. The section is too long and does not relate to main body of the technical description of the array and data.

**Authors: Well, shortening of this section is problematic, because it would be contradictory to the suggestions below (adding the discussion of the previous research), suggestions of the Rev.1 (adding the overview of the paper), and of the Rev. 3 (adding more detailed geological/tectonic background and justification of the importance of the research). We think these suggestions are justified and will improve the ms., so we decided to add the recommended parts. Anyway, we attempted to shorten remaining parts of the Introduction, where possible. We hope that it is acceptable in its present form.**

Moreover, a figure with tectonics of the region (including terms used in the text) and its location within Europe (an inset) are missing in this part. Not everybody is familiar with tectonics of the region.

**A: OK, we added inset to tectonic Figure 3, renamed it to Figure 2, and we refer to it in this section.**

Part starting in Line 35 contains a lot of tectonic statements taken from literature, but without a single reference. This is unacceptable.

**A: OK, references added (and tectonic description enlarged, according to the Rev. 3 suggestions).**

Moreover, the Introduction should mentioned previous studies devoted to the crust (a series of active experiments) or concerning to the mantle lithosphere (passive experiments), incl. anisotropy and LAB, sub-lithospheric upper mantle.

**A: OK. Originally, we presented only a very brief description of geological background and did not describe the previous research at all, because we felt that discussion of these topics in detail may not fit a largely technically-oriented paper. Now, according to the suggestions, we added information about previous studies (and geology). Also, we added suitable references as recommended. However, we do not agree with your next statement, and we explain it below:**

This is not the first study of this kind in the region as it is presented here

**A: Yes - it is not the first study, and No - we did not present our study as the first one in this region: In the second sentence of the ms. we called the region (=Polish Sudetes, as defined earlier) 'sparsely sampled' (which I guess is true when compared to well studied, Czech parts of BM). And, in this sentence, with citation (Wilde-Piórko et al., 2008) we did acknowledge PASSEQ 2006-2008 project as previously studying this area. 'Sparsely sampled' refers to the fact that the area of our present experiment was covered by not more than 3-4 broadband PASSEQ stations only, and also to the fact that numerous other seismic experiments studying the BM (e.g., BOHEMA II) were located outside this area - the Polish Sudetes (of course, at the mantle level, they partially cover this area as well, as the horizontal range of piercing points of teleseismic rays is broader than the extent of the network on the surface).**

**Obviously, these studies of the neighboring parts of the Bohemian Massif are also related to the subject of present project, so in the 'previous studies' paragraph we added appropriate citations.**

(e.g., Majdanski et al., 2014; Grad et al., 2008; Karousova et al., 2012; Geissler et al., 2012, Plomerova et al., 2012; Vecey et al., 2014; Knapmeyer-Endrun etal., 2013; Kind et al., 2017; Kvapil et al., 2021), regardless of the fact that the main purpose of the ms. lies in the AniMaLS array documentation.

**A: Yes, these and some other references were added.**

The second general comment relates to the repeating of too many numbers within the text, which are also in Figures, their captions and Tables. The continuous repeating makes the text chaotic and boring. Suppressing repetitions of several kinds will make the text shorter and clearly oriented on the technical aspects of the array.

**A: OK, we attempted to remove the repetitions.**

What is the reason for plotting in Fig. 1 also stations not used in the study and the figure at all?

**A: OK, we removed permanent stations not used in the experiment from figures 1, 3. However, we would prefer to keep in Fig. 1 the short period stations SUD1-6 and LUMINEOS stations. We do not describe them in detail here, because they are of secondary importance, but their recordings may be used in the future as complementary data.**

**The reason for Fig. 1 at all – we introduced this figure (even if locations of the stations are also in Fig. 3) to present technical parameters of the stations, topography (and in present, updated form, also geographical names used in the ms.) because putting all this information together with tectonic background, in a single figure, would, most likely, make it hard to read.**

Figure 3 would be more appropriate at the beginning of the ms.

**A: OK, Fig. 3 is now Fig. 2, and we refer to it at the beginning of the ms., just after the Fig. 1 (in the Introduction).**

According to which criteria some of permanent stations are listed in Table 1 (station used in the study, as seems form further parts in the text) and some not (see Fig. 1). The Table 1 can contain, e.g., the institutions operating/owning the stations instead in the long text.

**A: OK, we removed unused permanent stations from figures, so they are now consistent with Table 1. We added network names to the table. We think that adding institutions to the table is not necessary, as the institutions/full network names are already mentioned in the text.**

Not all abbreviations used in the text are introduced.

**A: OK, abbreviations are now explained.**

Line 18: strange formulation, usually data are used to model some feature (structure) .... not data Reformulate please.

**A: OK, reformulated.**

Line 39: should be …aims at

**A: OK, corrected.**

Line 67, 415: should be ….. Czech Regional Seismic Network

**A: OK, corrected.**

Line 179 (e.g.): consider to use …. data acquisition system…. insted of …..recorder

**A: OK, corrected in several places in the ms.**

Line 200: Cenozoic

**A: OK, corrected.**

Line 211: Sentence reformulation can help. Change order in the sentence, add (Fig.4) into it and delete the following two short sentences. Info in the second one is in the legend of the figure, there is no need to repeat it. Check the ms. for this kind of its improvement. It can be done in many places.

**A: OK, reformulated.**

Figures 6 and 7 are examples of regional and teleseismic earthquakes recorded by the network. Most information is repeated three times - in the figures, captions and text. Clean the text (line 230 and further) from numbers, keep them in figure headings/legends.

**A: OK, we removed redundant information from the text.**

Location of time scale at the top of figures with an arbitrary origin is strange.

**A: The origin (0) of the time axis is not arbitrary, it corresponds to the origin time of the event.**

Line 245: adding location of the two local earthquakes into the Fig. 3 (even if it becomes Fig.1) would bring new information.

**A: OK, location of the local event in the area of LGCD was marked on the map (Fig. 1). The location of the second event (Upper Silesia) is outside of both maps, unfortunately.**

Line 250: Do not repeat numbers presented elsewhere......The first event with magnitude 4.4 occurred 2018-07-03, 19:38:47.75 UTC …..

**A: OK, we removed redundant information from the text.**

Line 259 or caption of Fig. 10 and many other places in the text: who is using? Instead consider ... with the use of ....

**A: OK, corrected in the whole ms.**

PPSDs shown in Fig. 11 do not support the frequent statements on "high-quality data" from the AniMaLS array. AG10 does not meet the High and Low-noise criteria neither on the Z component. Horizontal components of AR06 are poor. Splitting of the PPSD at UPS (permanent) comes from the winter increase of the micro-seisms.

**A: OK, we removed "good-quality" statements from the ms.**

Showing PPSD for all three components of the all temporary stations (in an electronic attachment) would be better than Figure 12 with selected temporary stations and permanent stations, whose PPSD are better.

**A: OK, we prepared an extended version of Figure 12 (PPSDs of Z-comp. for all stations) for the electronic supplement. However, we would prefer not to remove the Fig. 12 from the ms.**

Is the effect of instrument type stronger than that of sediments at a site? It looks like that from Fig. 13. Please, comment on that. (Repetition of colour description in the legend and caption of Fig.13.)

**A: Yes, it seems that in the long period range, effect of instrument type is stronger – Guralp CMG-6T sensors consistently show high noise in this band, unlike the remaining sensors. We already commented on it at L. 296-300 of the original ms. (now the discussion is modified).**

Line 315: cultural effects – anthropogenic effects are meant (not necessarily "cultural")

**A: OK, corrected.**

Line 321: 3-component spectral seismograms

**A: OK, corrected.**

Line 333: Such phenomenon is well known for a long time and described .....

**A: OK, corrected.**

Line 340 and around: too many abbreviations

**A: OK, abbreviations expanded.**

Line 344: ….. corresponds to a thinning of the sedimentary layer, or,….

**A: OK, corrected.**

Line 356: repeats …

**A: OK, repetition removed.**

Line 364: change order in the sentence

**A: OK, done.**

Line 405 and Figure 15: change label of GKP to be readable; put all labels ABOVE the frame;

**A: OK, done.**

KEEP standard notation, i.e., that sensor mis-orientations greater than 5° prevents naming the horizontal components as N and E!!!, but the component must be denoted as Z, 1 (for N) and 2 (for E). This criterion should be accepted and followed (vs. Lines 412 and 430).

**A: This comment is a bit unclear for us, and we are not sure what are your recommendations and how should we proceed with corrections, therefore we would like to kindly ask for clarification of your point:**

**REV2: *'…KEEP standard notation, i.e., that sensor mis-orientations greater than 5° prevents naming the horizontal components as N and E!!!, but the component must be denoted as Z, 1 (for N) and 2 (for E)...'***

**A: We understand the convention for Z12 naming of components, instead of ZNE naming, if (documented) mis-orientation exceeds 5°. But, in discussion in this Section we do not explicitly refer to components, so we are not sure where exactly should we use this convention.**

**REV. 2: *'…This criterion should be accepted and followed (vs. Lines 412 and 430).'***

**A: In these lines, we mention that most stations do not exceed -7° to + 7° mis-orientation in results from polarization methods, and that only the 3 stations exceeding 10° will be rotated before using for analyses, because polarization methods show substantial uncertainty and lower values are not a good proof of mis-orientation.**

**If we get your point right, you recommend to rotate not only the three stations exceeding 10°, but, consistently with convention, to rotate all stations exceeding 5° calculated with polarization analysis.**

**However, in our opinion, this recommendation would apply only to the situation when the mis-orientations are determined with high precision, directly in the field. Given substantial uncertainty of the polarization analysis methods (estimated errors of the three methods used are largely 4-7° in Table 2, Vecsey et al. (2017) mention even larger (10°) error as typical for these methods), results showing 5° or 7° values based on the polarization analysis are not a definitive proof of mis-orientation and, in this case, orientations determined in field using method described in the paper should be trusted, as more precise.**

**We would be grateful for your more detailed comment on these issues, we are ready to introduce additional corrections to the manuscript according to your suggestions.**

Wrong orientation of the GKP station has been identified during processing data from the PASSEQ experiment and published in Vecsey et al. (2014)! However, presenting this information in the ms. is not correct, or, written in confusing way.

**A: Yes, we know about Vecsey et al. (2014) result, in the orig. ms. we referred to it: *'…station GKP is outside the study area, but it is shown for comparison, as previous studies also reported its significant misorientation - Vecsey et al. (2014) reported 41°, Wilde-Piórko et al. (2017) reported 39° and 45°, result of this study: 34-37°.'***

**Now, according to the reviewer's suggestions, we also added these citations to Fig. 15 caption and to the Table 2.**

What do you mean under the direct measurements in the field? In this form I understand that you mean measurements by gyrocompass.

**A: Yes.**

However, the GKP mis-orientation in Vecsey et al. (2014) was determined by software methods, as described there and mentioned in the Figure 15 captions (add ref Vecsey et al., 2014 for GKP deviation) but wrongly in the rightmost column of Table 2. I suggest:

Table 2. Misorientation angles for all stations by different methods.

| Station Code | Fontaine (2009) | BNG method | DL method | other / gyrocompass |
|---|---|---|---|---|
| AG01 | | | | |

...................
GKP     34.36+/-          36.80+/- Â     33.92+/- Â     39+/- (Wilde-Piorko etal, 2017)
                                    45+/- (Wilde-Piorko etal, 2017)
                                    41 (Vecsey et al., 2014)
Followed by as you have it
CHVC 11.07 Â  } 4.64 8.61 Â  } 5.50 6.50 Â  } 4.82 0.70
DPC -0.52 Â  } 2.72 1.60 Â  } 2.66 -2.27 Â  } 3.00 0.00
KRLC -1.72 Â  } 3.16 -1.22 Â  } 2.38 -1.33 Â  } 4.33 0.00
MORC -2.57 Â  } 3.09 -1.91 Â  } 3.21 -0.52 Â  } 3.10 --
OKC -8.68 Â  } 3.09 -11.03 Â  } 3.16 -5.99 Â  } 3.62 -7.00
OSTC 21.73 Â  } 4.33 28.26 Â  } 6.82 21.84 Â  } 3.59 23.00
UPC -0.05 Â  } 3.55 4.13 Â  } 4.35 -0.53 Â  } 3.06 0.00

**A: OK, we modified Table 2 and added the citations,  citations added also to Fig. 15 caption.**

Line 416: Modify the sentence  ……from direct, high accuracy measurements in field by gyrocompass (Vecsey L., Institute of Geophysics of Czech Academy of Sciences, personal communication, 2020).

**A: OK, done.**

Line 453: correct the sentence and add references for active experiments.

**A: This part of the text is now removed according to the Rev.1 suggestions, but references to active experiments were added to the Introduction section.**

Line 455: In the past, the study area was covered by a single Polish seismic station KSP only …… This is incorrect statement. See, e.g, Fig. 2 in Vecsey et al., 2014 for PASSEQ experiment and other experiments covering the Sudetes (BOHEMA II) and related papers.

**A: Yes,  the sentence  was incorrect and is now removed. We wanted to comment the distribution of permanent stations, and to say that there was only one permanent station  in the Polish Sudetes, but it was incorrectly formulated and it got different meaning than intended. Indeed, the PASSEQ experiment also conducted measurements there, as we cited it in orig. manuscript (Wilde-Piórko et al., 2008),**

**(The remaining sentence of the paragraph was moved to the Introduction, according to the Rev. 1 suggestion to shorten the Conclusions section).**

Line 465: Data availability: Data from the AniMaLS experiment are stored at the IG PAS (https://dataportal.igf.edu.pl/dataset/animals), currently with restricted access. Do you plane to store the data in an EIDA node and when the data will be opened for public access?

**A: OK, we added the information about the data access – the data will be open in 2023. The data are now stored in the IG PAS data portal, perhaps in the future they will be stored in the EIDA, but we are not sure.**

---

## Author Comment (AC3)

**Dear Dr. Geissler,**

**We are grateful to have received your review of our manuscript which led to important improvements of the manuscript. Below we address your comments in detail. Our responses are written in bold font.**

The manuscript presents a passive seismic experiment carried out in the Polish Sudetes, an important region to understand the tectonic evolution of the Varscan Orogen in Central Europe. The authors present how a state-of-the-art seismological experiment was and can be carried out. Furthermore, the authors discuss the quality, the potential and limitations of the data set.

I have two major points that should be considered for a revision of the manuscript: 1) I miss a more clear, elaborated introduction into the geology and tectonics of the study area. Why it is important to study this region in more detail?

**A: OK, we added broader description of geology and tectonics, as well as motivation.**

2) Some of the figures need minor revisions, e.g, Figure 1 is missing more geographical or geological names. Maybe the geological map should be placed earlier in the manuscript. Furthermore, it is missing a legend.

**A: OK, geographical names were added to Fig. 1. We added the legend to Fig. 3 (geol. map), and this figure is renamed to be now the Fig. 2., so we can refer to it in the Introduction.**

All subfigures should have similar scale (e.g., Fig. 13).

**A: OK, we used the same scale, and according to the suggestion in the annotated pdf, beside the 'night' PPSDs version we added also 'day' PPSDs version for comparison. Discussion of the figure in the text is changed accordingly.**

Additionally, all abbreviations should be introduced (both in text and captions). Please rethink, if the term "Tertiary" is necessary to use. It is not an official term anymore.

**A: OK, abbreviations are now explained. "Tertiary" is removed.**

Minor comments, questions, suggestions you will find in the annotated manuscript.

**A: OK, we addressed all remarks from the annotated manuscript. Our explanations are added to the annotated manuscript, attached as a supplement to this discussion. Corrections accepted by us are mostly left unmarked or sometimes marked with green and with our comments in speech bubbles to clarify the point. Very few suggestions that we would prefer to decline are marked with dark yellow color, with our comments in speech bubbles in order to explain the reason.**

---

## Author Comment (AC4)

**Passive seismic experiment 'AniMaLS' in the Polish Sudetes (NE Variscides)**

Monika Bociarska1, Julia Rewers1, Dariusz Wójcik1, Weronika Materkowska1, Piotr Środa1 and *AniMaLS Working Group*\*

[revised manuscript text omitted]

25-30 km. 60

---

## Referee Report (RR1)

Review of revised version of ms. **Passive seismic experiment 'AniMaLS' in the Polish Sudetes (NE Variscides)** by M. Bociarska et al.

I am satisfied with the correction in the revised version of ms. gi-2021-7 and explanations in the authors reply. Only the misunderstanding with seismogram component notation ZNE vs. Z12 needs a short explanation. I do not insist on any rotation of the components, especially if the authors are not sure with the correct orientation of the seismometers (+/-2° is not much). I only recommended to follow the standard agreement that if misorientation exceeds 5°, the component should be noted Z12.  Discussion on how many stations exceed +/-7° or their misorientation is even larger than 10° relates only to a description of the network quality. After a certain time, the data will be accessible openly to a broad seismological community. Therefore, I only recommended follow the standard rules, which in fact warns the users. At any case I recommend using the gyrocompass for getting the seismometers oriented towards the North as accurate as possible during the installation as well as to check the orientation from time to time during the network operation.

Reviewer 2